# Generation of functional hepatocytes by forward programming with nuclear receptors

Rute A Tomaz[1,2], Ekaterini D Zacharis[1,2], Fabian Bachinger[1,2], Annabelle Wurmser[1,2], Daniel Yamamoto[1,2], Sandra Petrus-Reurer[2], Carola M Morell[1,2], Dominika Dziedzicka[1,2], Brandon T Wesley[1], Imbisaat Geti[1,2], Charis-Patricia Segeritz[1,2], Miguel C de Brito[1,2], Mariya Chhatriwala[2], Daniel Ortmann[1,2], Kourosh Saeb-Parsy[2], Ludovic Vallier[1,2,3]*

[1]Wellcome-MRC Cambridge Stem Cell Institute, University of Cambridge, Cambridge, United Kingdom; [2]Department of Surgery, University of Cambridge and NIHR Cambridge Biomedical Research Centre, Cambridge, United Kingdom; [3]Wellcome Sanger Institute, Wellcome Genome Campus, Hinxton, United Kingdom

**Abstract** Production of large quantities of hepatocytes remains a major challenge for a number of clinical applications in the biomedical field. Directed differentiation of human pluripotent stem cells (hPSCs) into hepatocyte-like cells (HLCs) provides an advantageous solution and a number of protocols have been developed for this purpose. However, these methods usually follow different steps of liver development in vitro, which is time consuming and requires complex culture conditions. In addition, HLCs lack the full repertoire of functionalities characterising primary hepatocytes. Here, we explore the interest of forward programming to generate hepatocytes from hPSCs and to bypass these limitations. This approach relies on the overexpression of three hepatocyte nuclear factors (*HNF1A*, *HNF6,* and *FOXA3*) in combination with different nuclear receptors expressed in the adult liver using the OPTi-OX platform. Forward programming allows for the rapid production of hepatocytes (FoP-Heps) with functional characteristics using a simplified process. We also uncovered that the overexpression of nuclear receptors such as RORc can enhance specific functionalities of FoP-Heps thereby validating its role in lipid/glucose metabolism. Together, our results show that forward programming could offer a versatile alternative to direct differentiation for generating hepatocytes in vitro.

*For correspondence:
lv225@cam.ac.uk

## Editor's evaluation

The work by Vallier and colleagues programmes ESCs and IPSCs towards hepatocyte fate by using a combination of hepatocyte transcription factors. Based on informatic analyses comparing adult hepatocytes with hepatocyte-like cells differentiated with soluble factors they conclude that the inclusion of RORc is important for added maturity of the forward programmed cells. The challenge is a very important one as we still don't have good in vitro hepatocyte generation.

## Introduction

Hepatocytes are the main cell type of the liver, comprising 80% of its volume and performing a vast array of vital functions including lipid metabolism, storage of macronutrients, secretion of plasma proteins, and xenobiotic detoxification (*Gordillo et al., 2015*; *Si-Tayeb et al., 2010*; *Trefts et al., 2017*). Diseases affecting these functions are life-threatening and end-stage forms require liver

transplantation. However, only a limited number of patients can benefit from this therapy due to scarcity of donors and the side effects of immunosuppression. Cell-based therapy using primary hepatocytes has already been found to be an attractive therapeutic alternative to whole organ transplants (*Dhawan et al., 2020*). However, primary human hepatocytes (PHHs) are in short supply as they can only be obtained from suboptimal livers unsuitable for transplantation. Furthermore, they display a short life, absence of proliferation, and rapid loss of functionality in vitro (*Mitry et al., 2002*). Similarly, the development of new platforms for drug development and toxicology screens is greatly affected by the lack of robust sources of hepatocytes. For all these reasons, alternative sources of hepatocytes are urgently needed. Producing hepatocytes from human pluripotent stem cells (hPSCs) using directed differentiation protocols has been shown to be an advantageous alternative to PHHs (*Palakkan et al., 2017*; *Szkolnicka and Hay, 2016*). These protocols commonly follow key stages of liver development in vitro and allow the production of hepatocyte-like cells (HLCs) which exhibit key hepatic functions including Albumin secretion, lipid metabolism, glycogen storage, and urea cycle activity. However, HLCs systematically present an immature/foetal-like phenotype lacking the full repertoire of functions of mature hepatocytes (*Baxter et al., 2015*; *Grandy et al., 2019*; *Yiangou et al., 2018*). The development of fully functional hepatocytes in vitro is challenging due to the lack of detailed knowledge concerning the molecular mechanisms driving functional maturation in vivo. Maturation is a process that occurs progressively , and mimicking this timeline and the associated combination of metabolic changes, exposure to oxygen, nutrition, and microbiome constitutes a major challenge for direct differentiation protocols (*Chen et al., 2011*). As an alternative, overexpression of transcription factors (TFs) has been explored as a way to improve functionality of in vitro generated hepatocytes (*Boon et al., 2020*; *Nakamori et al., 2016*; *Zhao et al., 2013*). Moreover, transdifferentiation of somatic cells into liver cells has been achieved by overexpression of liver-enriched transcription factors (LETFs) in mouse and human fibroblasts (*Rombaut et al., 2021*). Importantly, these LETFs comprise the HNF1, HNF3 (FOXA), HNF4, and HNF6 (ONECUT) families all of which play key roles in coordinating liver development (*Gordillo et al., 2015*; *Lau et al., 2018*; *Schrem et al., 2002*). However, direct cell conversion from somatic cell types has a low efficiency/yield due to the strong epigenetic restrictions present in fully differentiated cells. Furthermore, somatic cells have restricted capacity of proliferation which limits large-scale production of hepatocytes without the use of oncogenic manipulation (*Du et al., 2014*; *Huang et al., 2014*). Forward programming by direct overexpression of TFs in hPSCs could bypass these limitations. Indeed, the epigenetic state of hPSCs is more permissive to direct cellular conversion, while their capacity of proliferation is far more superior to somatic cells. Accordingly, this approach has been successfully used to generate neurons, skeletal myocytes, and oligodendrocytes by taking advantage of the OPTi-OX system (*Pawlowski et al., 2017*). Here, we decided to exploit the same platform to produce hepatocytes by forward programming. We first tested different combinations of LETFs and identified a cocktail of three factors sufficient to drive the conversion into hepatocytes. We then performed transcriptomic and epigenetic comparisons between HLCs and PHHs to identify additional TFs which could further increase the functional maturation of hepatocytes. This comparison revealed that a number of nuclear receptors are expressed in adult hepatocytes and thus are likely to be inducers of functionality and maturation in vivo. A selection of these factors was combined with LETFs and we identified that the 4TFs HNF1A-HNF6-FOXA3-RORc were the most efficient cocktail to generate hepatocytes by forward programming (FoP-Heps) displaying features of mature hepatocytes including CYP3A4 activity, protein secretion, and hepatotoxic response. Thus, forward programming offers an alternative to direct differentiation, bypassing the need for complex culture conditions and lengthy timelines. Moreover, FoP-Heps display a level of functionally relevant for regenerative medicine, as well as disease modelling or drug screening.

## Results
### LETFs allow forward programming into cells with hepatocyte identity

The first step to develop a forward programming method consists in identifying a cocktail of TFs which can recreate the transcriptional network characterising the target cell type. However, this step is challenging for hepatocytes as liver development is not initiated by a single and specific master regulator, and the factors driving functional maturation of hepatocytes remain to be fully uncovered. To bypass these limitations, we decided to focus on the LETFs which are known to control the induction

of the hepatic program during foetal development and have been tested in somatic cell conversion (*Rombaut et al., 2021*). The coding sequence of four LETFs (*HNF4A, HNF1A, HNF6,* and *FOXA3*) was cloned into the OPTi-OX system (*Figure 1A*) and the resulting inducible cassette was targeted into the AAVS1 gene safe harbour (*Bertero et al., 2016*; *Pawlowski et al., 2017*). After selection, individual sublines were picked, expanded, and genotyped before further characterisation. Addition of doxycycline (dox) for 24 hr was sufficient to induce homogenous and robust expression of each LETF in the selected human embryonic stem cells (hESCs) (*Figure 1B and C*) confirming the efficacy of the OPTi-OX system in inducing transgene expression. Importantly, this induction was not associated with differentiation into liver cells (data not shown) suggesting that LETFs alone are not sufficient to impose an hepatocytic identity. Thus, we decided to screen culture conditions which could sustain both the survival and differentiation of hepatocytes (data not shown) and found that after the initial 24 hr in E6 medium, the cells acquired a hepatocyte-like morphology when cultured in Hepatozyme complete medium for 14 days (*Figure 1D*). Interestingly, the resulting cells expressed hepatocyte markers such as Albumin (*ALB*), Alpha-1 Antitrypsin (*A1AT* or *SERPINA1*), and Alpha-Fetoprotein (*AFP*) (*Figure 1E*) and displayed CYP3A4 activity levels comparable to HLCs generated by direct differentiation, albeit several fold lower than PHHs (*Figure 1F*). Next, we asked whether all the four LETFs were necessary to achieve this hepatocyte-like phenotype. For that, we removed each factor to generate hESCs sublines expressing combinations of three factors (*Figure 1—figure supplement 1A*). Robust and homogeneous expression at the protein level was again confirmed after 24 hr of dox induction (*Figure 1—figure supplement 1B*). Induction of each combination of three LETFs in culture conditions identified above showed that *HNF1A, HNF6,* or *FOXA3* were necessary to generate cells expressing hepatocytes markers such as *ALB,* albeit with heterogeneity at the protein level (*Figure 1G*, *Figure 1—figure supplement 2A,B*). *HNF4A* overexpression seemed to be dispensable as cells generated by overexpression of the three remaining LETFs (*HNF1A, HNF6, FOXA3*) acquired a cobblestone-like morphology, and expressed high levels of *ALB, SERPINA1,* and *AFP* (*Figure 1G, H and I*, *Figure 1—figure supplement 2*). Strikingly, hepatocytes generated using these 3TFs (3TF FoP-Heps) achieved the highest levels of CYP3A4 activity suggesting overexpression of *HNF4A* itself is unnecessary to acquire the hepatocyte characteristics screened, as its expression might be induced by one of the three LETFs (*Figure 1J*). Altogether, these results showed that overexpression of *HNF1A, HNF6,* and *FOXA3* is sufficient to forward program hPSCs towards HLCs.

## HLCs generated by direct differentiation lack the expression of specific nuclear receptors

Following these encouraging results, we aimed to increase the functionality of 3TF FoP-Heps by adding TFs which could play a role in promoting hepatic maturation and functionality. However, identifying these factors proved to be challenging as there is little information about the mechanisms driving functional maturation of hepatocytes, especially following birth when adult hepatic functions are established. To bypass this limitation, we decided to compare the transcriptome profile of adult PHHs to the transcriptome of HLCs generated from hPSCs by direct differentiation. Indeed, HLCs represent a foetal state which has been broadly characterised (*Baxter et al., 2015*), while 3TFs FoP-Heps are likely to be less relevant for natural development. For this comparison, we used a state-of-the-art protocol (*Hannan et al., 2013*; *Touboul et al., 2010*) which has been used for modelling liver disease (*Rashid et al., 2010*; *Segeritz et al., 2018*) and as proof of concept for cell-based therapy applications (*Yusa et al., 2011*). This protocol starts by the production of endoderm cells expressing *SOX17*, followed by the specification of foregut expressing *HHEX*, after which cells transit through a hepatoblast-like state marked by *TBX3* (*Figure 2—figure supplement 1A*). Interestingly, LETFs were expressed during this differentiation protocol at levels comparable to PHHs (*Figure 2—figure supplement 1B,C*) confirming that these first steps follow a natural path of development. The resulting progenitors undergo a final stage of differentiation into HLCs expressing functional markers such as *ALB* and *SERPINA1* (*Figure 2A*, *Figure 1—figure supplement 1A,D*). Despite displaying key hepatic functions (*Baxter et al., 2015*; *Grandy et al., 2019*; *Yiangou et al., 2018*), HLCs represent a 'foetal' state as shown by the expression of *AFP* (*Figure 2—figure supplement 1A,D*) or by the limited activity/expression of *CYP3A4, CYP2A6,* or *CYP2C9* (*Figure 1B*, *Figure 2—figure supplement 1E*). RNA-sequencing (RNA-seq) performed on HLCs generated from either human induced pluripotent stem cells (hiPSCs) or hESCs, and from freshly harvested PHHs (fPHHs) or cultured in vitro

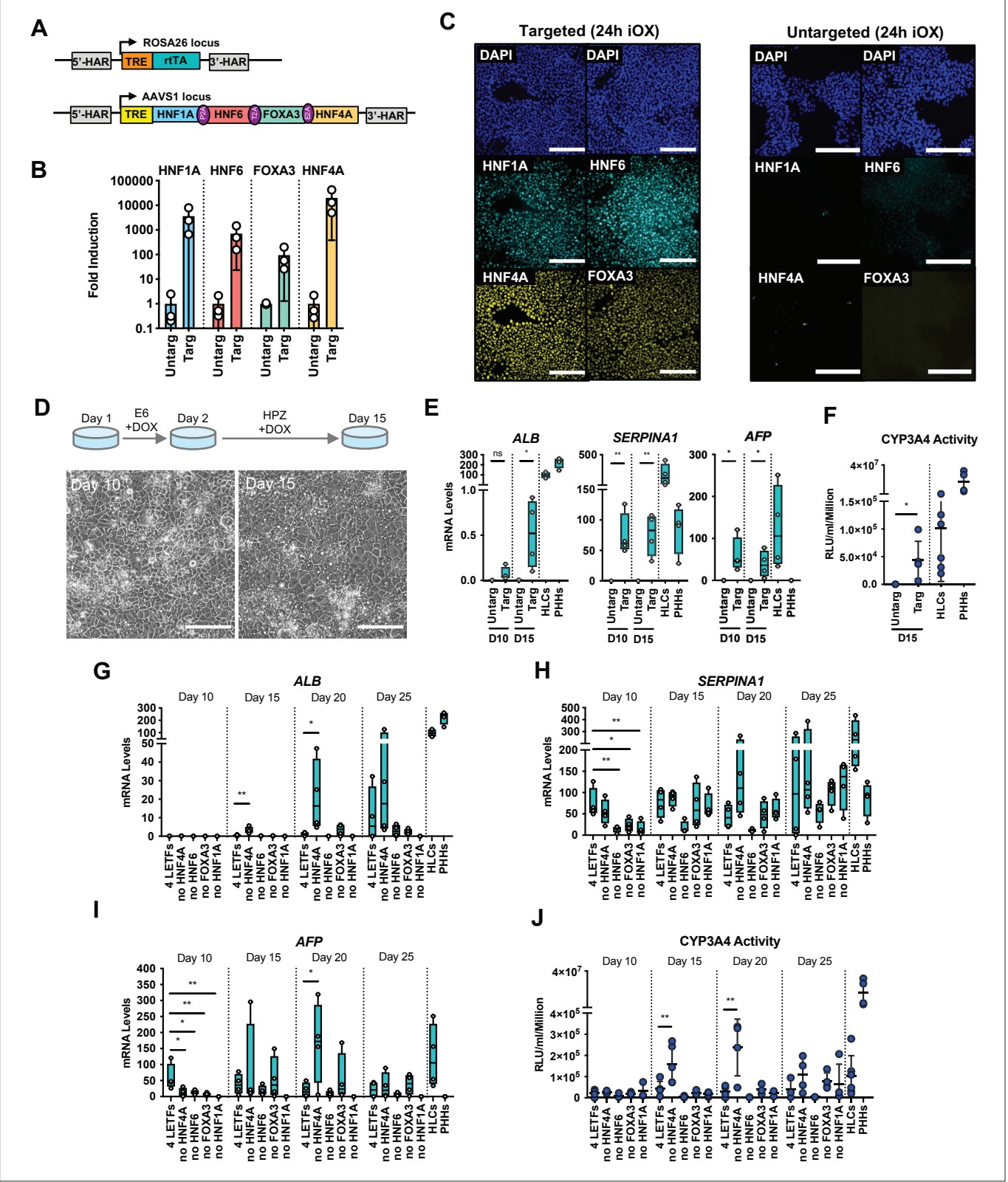

**Figure 1.** Forward programming of human pluripotent stem cells (hPSCs) into hepatocytes with four and three liver-enriched transcription factors (LETFs). (**A**) Schematic representation of the two sequentially targeted loci. The human ROSA26 was targeted with a constitutively expressed reverse tetracycline transactivator (rtTA). The AAVS1 locus was targeted with the four LETFs *HNF1A*, *HNF6*, *FOXA3*, and *HNF4A* downstream of a TET-responsive element (TET). (**B**) mRNA induction levels of the four factors in targeted human embryonic stem cells (hESCs) (Targ) relative to untargeted

*Figure 1 continued on next page*

*Figure 1 continued*

(Untarg) hESCs stimulated with doxycycline (dox) for 24 hr (n=3). Data is shown relative to the untargeted control. (**C**) Immunofluorescence staining of the four LETFs in targeted and untargeted hESCs after 24 hr of inducible overexpression (iOX) with dox confirming transgene induction. Nuclei were counterstained with DAPI (blue). Scale bar, 200µm. (**D**) Schematic representation of the iOX culture conditions for forward programming. Phase contrast images of hESCs targeted with the four LETFs after 10 and 15 days of forward programming. Scale bar, 200µm. (**E**) mRNA levels of hepatocyte markers (*ALB*, *SERPINA1*, and *AFP*) in hESCs targeted with the four LETFs after 10 and 15 days of forward programming. Untargeted hESCs treated with the same protocol as in (**D**) were used as control (n=4). Statistical difference was calculated with unpaired t-test against untargeted. (**F**) CYP3A4 activity levels normalised per cell number (millions) in untargeted and targeted hESCs with the four LETFs after 15 days of forward programming (n=5) . Statistical difference between targeted and untargeted cells was calculated with unpaired t-test. (**G,H,I**) mRNA levels of hepatocyte markers (*ALB*, *SERPINA1*, and *AFP*) in hESCs targeted with the four LETFs and with combinations of three LETFs (n=4). The factor removed from each construct is indicated. Expression levels were determined after 10, 15, 20, and 25 days of forward programming. Statistical differences were calculated with one-way ANOVA, corrected for multiple comparisons compared to four LETFs. All mRNA levels were normalised to the average of two housekeeping genes (*PBGD* and *RPLP0*). (**J**) CYP3A4 activity levels normalised per cell number (millions) in hESCs targeted with the four LETFs and combinations of three LETFs after 10, 15, 20, and 25 days of forward programming (n=3–5). Statistical differences were calculated with one-way ANOVA, corrected for multiple comparisons compared to four LETFs. In all plots, bars represent mean with SD, and individual datapoints are shown for all biological replicates. Hepatocyte-like cells (HLCs) generated by direct differentiation and primary human hepatocytes (PHHs) where plotted as controls for all CYP3A4 activity and expression data. Significant p-values are shown at each comparison and indicated as *p<0.05, **p<0.01, ***p<0.001, ****p<0.0001.

The online version of this article includes the following source data and figure supplement(s) for figure 1:

**Source data 1.** Individual measurements and statistical tests related to *Figure 1*.

**Figure supplement 1.** Validation of inducible overexpression (iOX).

**Figure supplement 2.** Characterisation of the phenotype of human pluripotent stem cells (hPSCs) forward programmed into hepatocytes with three liver-enriched transcription factors (LETFs).

as monolayer (pPHHs) reinforced these observations. Principal component analysis (PCA) of the most variable 500 genes showed a clear distinction between the three cell types, with HLCs clustering in-between undifferentiated hiPSCs and PHHs confirming their intermediate state of differentiation (PC1: 52%, *Figure 2C*). In order to further explore the differences between HLCs and PHHs, we combined differential gene expression (DGE) and gene ontology (GO) analyses to identify genes and biological functions specific to each cell type (*Figure 2D*). Genes uniquely expressed in PHHs (cluster 1) were associated with adult liver functions such as response to xenobiotic and xenobiotic metabolism, inflammatory response, and complement activation (*Figure 2E*). Genes expressed in both HLCs and PHHs (Heps; cluster 3) were involved in liver development, fatty acid metabolism, or broad cellular functions (*Figure 2F*). Of note, genes specifically upregulated in HLCs (cluster 2) were associated with extracellular matrix organisation and varied developmental functions which could originate from their in vitro environment (*Figure 2—figure supplement 1F*). We then decided to focus specifically on TFs using a previously curated list (*Lambert et al., 2018*; *Figure 2G and H*) and identified 36 TFs highly expressed in PHHs vs. HLCs (p<0.05, log2 fold change >2). Interestingly, reactome pathway analysis grouped these TFs into two main pathways: the NFI family and a cohort of eight nuclear receptors (*Figure 2H*). Nuclear receptors are known to be involved in key liver functions including the metabolism of lipids and glucose, bile acid clearance, xenobiotic sensing, and regeneration (*Rudraiah et al., 2016*). Thus, we hypothesise that nuclear receptors could be the most promising candidates to improve hepatocyte functionality in our culture system. Taken together these observations show that the immature state of HLCs is associated with the absence of several nuclear receptors thereby suggesting that these factors could be necessary to drive functional maturation of hepatocytes.

## Epigenetic characterisation of HLCs suggests a role for nuclear receptors RORc, AR, and ERα

To further refine the list of nuclear receptors identified by our transcriptomic analyses, we decided to compare the epigenetic landscape of HLCs vs. PHHs. Indeed, we hypothesised that nuclear receptors binding regulatory regions in PHHs could have a key function in maturation and thus we aimed to identify the most important factors by screening underlying motifs in such regions. Chromatin immunoprecipitation-sequencing (ChIP-seq) was performed on histone marks including H3K27ac (active regulatory regions), H3K4me1 (active or primed regulatory regions), and H3K27me3 (silenced genes) (*Creyghton et al., 2010*; *Wang et al., 2015*). These marks were profiled in HLCs derived from both hiPSCs and hESCs, and PHHs while undifferentiated hiPSCs were used as control. As

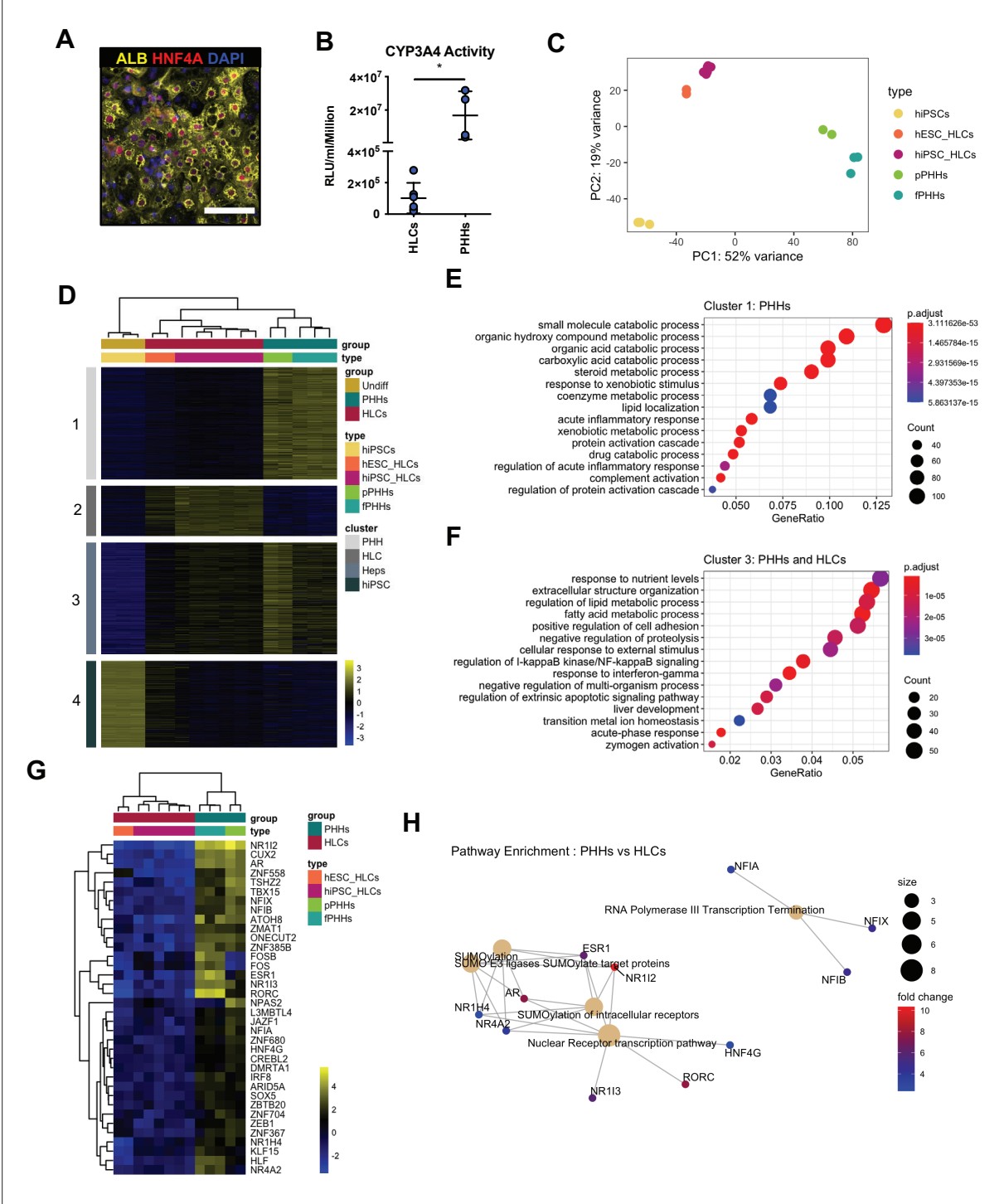

**Figure 2.** Hepatocyte-like cells (HLCs) and primary human hepatocytes (PHHs) display transcriptomic differences associated with their state of maturation. (**A**) Immunofluorescence staining of Albumin (yellow) and HNF4A (red) in HLCs differentiated for 30 days. Nuclei were counterstained with DAPI (blue). Scale bar, 100 μm. (**B**) CYP3A4 activity levels normalised per cell number (millions) in HLCs differentiated for 30 days (n=6) and PHHs (n=4). Bars represent mean with SD, and individual datapoints represent the different biological replicates. Statistical difference was calculated with unpaired t-test. (**C**) Principal component analysis (PCA) of undifferentiated human induced pluripotent stem cells (hiPSCs), HLCs derived from human embryonic stem cell (hESC) (hESC_HLCs) and hiPSC (hiPSC_HLCs), freshly harvested PHHs (fPHHs), or plated PHHs (pPHHs). (**D**) Heatmap showing the proportion of genes differentially expressed in each cell type (cluster 1 – PHHs, cluster 2 – HLCs, cluster 4 – hiPSCs) as well as in Heps (HLCs and PHHs) against undifferentiated hiPSCs (cluster 3). (**E, F**) Dotplot showing the top 15 hits on gene ontology enrichment analysis on genes associated to cluster 1 and cluster 3 as shown in (**D**). The size of each dot represents number of genes associated to each term and the colours represents the adjusted p-value.

*Figure 2 continued on next page*

Figure 2 continued

(**G**) Heatmap showing the differential gene expression of transcription factors between PHHs (fresh or plated) and HLCs (hESC and hiPSC derived).

(**H**) Reactome pathway enrichment analysis on transcription factors identified in (**G**). Differential gene expression was calculated with log2(fold change) higher than 2 and adjusted p-value <0.05. Hierarchical clustering on samples was generated by Euclidean distance.

The online version of this article includes the following source data and figure supplement(s) for figure 2:

**Source data 1.** Individual measurements and statistical tests related to *Figure 2* and corresponding supplements.

**Source data 2.** List of genes differentially expressed in the four clusters in *Figure 2D*.

**Figure supplement 1.** Characterisation of transcriptional differences between hepatocyte-like cells (HLCs) and primary human hepatocytes (PHHs).

expected, PCAs showed a marked divergence between the epigenetic profile of HLCs and hiPSCs independently of the mark analysed (*Figure 3A*). Interestingly, HLCs and PHHs clustered in close proximity suggesting that these cell types share an important part of their epigenetic profile despite their transcriptomic differences. The profiles of H3K27ac and H3K27me3 showed the highest variance between HLCs/PHHs and hiPSCs, confirming the importance of these marks for establishing cellular identity (*Figure 3A*). We then performed differential peak calling on H3K27ac to identify regulatory regions uniquely enriched and active in PHHs vs. HLCs ('PHH-specific'), and vice versa ('HLC-specific') (*Figure 3—figure supplement 1A*). We also profiled H3K4me1 and H3K27me3 in either PHH or HLC-specific regions. This analysis revealed that H3K4me1 was absent at 'PHH-specific' regions in HLCs and seems to be broadly replaced by spread of H3K27me3 deposition instead (*Figure 3B*). Interestingly, discrete portions of regulatory regions lacked H3K27ac in genes downregulated in HLCs (see for example *CYP3A4* and *UGT1A*, *Figure 3C*, *Figure 3—figure supplement 1B*). Ontology of genes associated with each set of regions highlighted several adult liver metabolic processes in the 'PHH-specific' set such as steroid, lipid, and xenobiotic metabolism, and range of neural functions for 'HLC-unique' regions (*Figure 3—figure supplement 1C*), in agreement with the transcriptomic analyses. Overall, these results suggested that a subset of genes involved in adult liver functions lack H3K4me1 priming as well as full H3K27ac deposition in HLCs. These regions lacking H3K27ac in HLCs appeared to display repressive marks such as H3K27me3. In addition, HLCs displayed active histone marks in regions including genes which are not associated with liver differentiation confirming that cells generated from hPSCs also present an epigenetic signature specific to their in vitro state (*Figure 3—figure supplement 1C*). Taken together these observations suggested that HLCs and PHHs broadly share the same epigenetic identity when compared to hiPSCs. However, the activation of a limited and specific set of regulatory regions is missing in HLCs, which could explain the absence of expression of adult liver genes and lack of functional maturation. To identify the nuclear receptors potentially involved in regulating these regions, we performed motif enrichment analysis in the 'PHH-specific' regions marked by H3K27ac. Interestingly, this analysis identified a significant enrichment for the androgen (AR) and estrogen (ERα) response elements, as well as RORc motifs (*Figure 3D*), which were among the top differentially expressed nuclear receptors in our transcriptomic analyses. We then decided to further investigate the importance of these nuclear receptors throughout development using mouse RNA-seq datasets obtained at different stages of liver organogenesis E12.5, E16.5, P0, 8-week and 10-week adults (*Figure 3—figure supplement 1D*). Interestingly, the expression of these three nuclear receptors was found to be upregulated specifically in the adult liver (*Figure 3—figure supplement 1E,F*). Altogether, these observations suggested that the nuclear receptors AR, ERα, and RORc could play a role in establishing or maintaining a transcriptional network characterising mature hepatocytes.

## Overexpression of RORc increases the functionality of hepatocytes generated by forward programming

We next tested the capacity of RORc (*RORγ*), AR, and ERα (*ESR1*), to further improve the functionality of FoP-Heps generated using three LETFS. For that, we generated hESC lines inducible for the expression of HNF1A, HNF6, and FOXA3 (3TFs) in combination with each of the nuclear receptors identified above (*Figure 4—figure supplement 1A*). The homogeneous induction of the 4TFs was validated using immunostaining. Interestingly, these analyses showed that the overexpressed nuclear receptors were located in the nucleus, and in the case of AR in both cytoplasm and nucleus (*Figure 4—figure supplement 1B,C,D*). We then induced forward programming using the culture

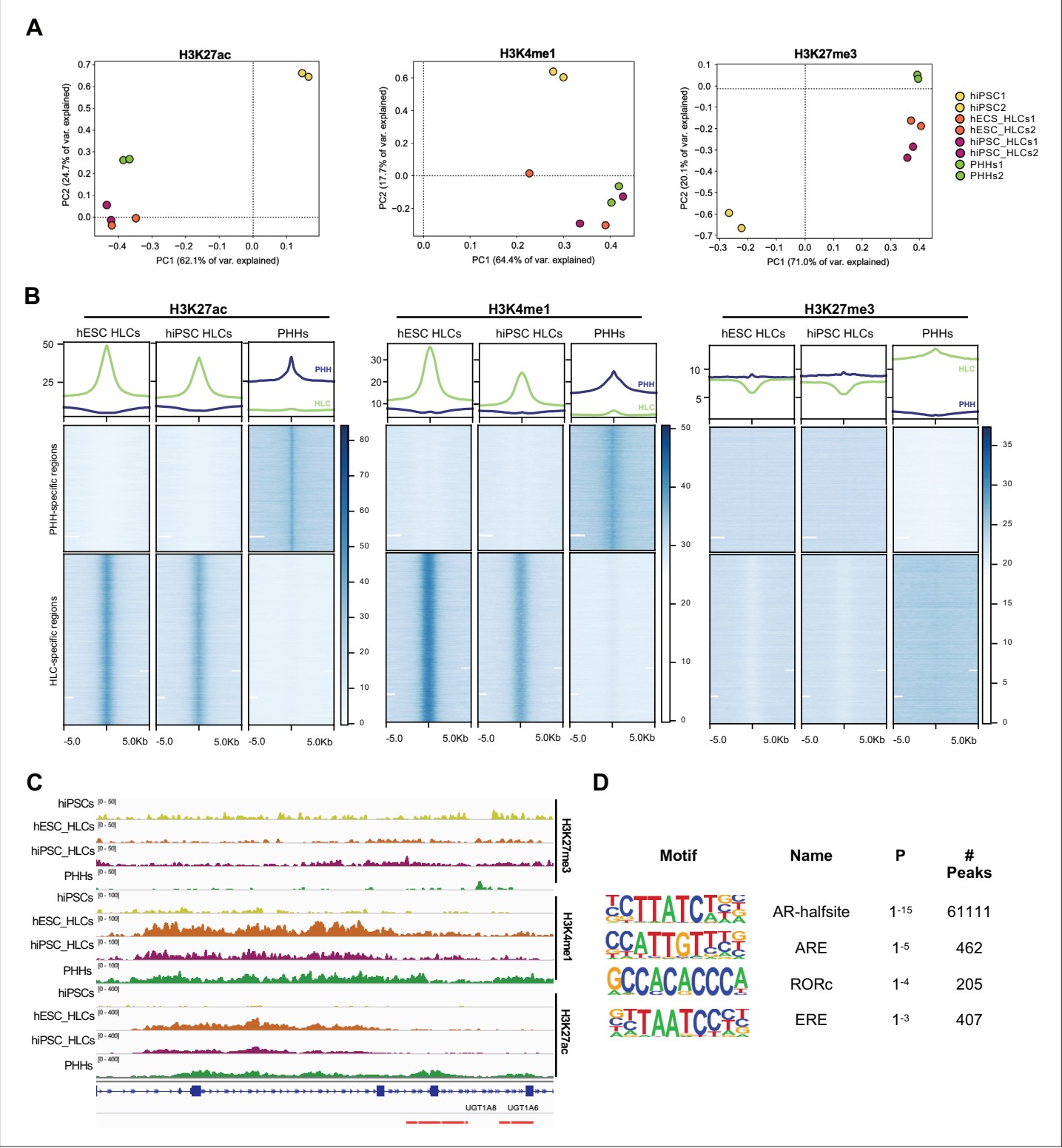

**Figure 3.** Epigenetic status of regulatory regions differs between states of maturation in hepatocyte-like cells (HLCs) and primary human hepatocytes (PHHs). (**A**) Principal component analysis (PCA) of the global enrichment profile of H3K27ac, H3K4me1, and H3K27me3 across two replicates of undifferentiated human induced pluripotent stem cells (hiPSCs), human embryonic stem cell (hESC), and hiPSC-derived HLCs, and PHHs. Average scores were computed for genomic regions of 1000 bp for the entire genome. (**B**) Average density plots and heatmaps showing enrichment levels for H3K27ac, H3K4me1, and H3K27me3 within a 10 kb window centred at H3K27ac PHH-unique (blue) or HLC-unique (green) regions. Scales are adjusted to maximum peak intensity for each dataset. (**C**) Enrichment profiles of H3K27ac, H3K4me1, and H3K27me3 across the *UGT1A* locus. Profiles are shown

*Figure 3 continued on next page*

*Figure 3 continued*

for one replicate of undifferentiated hiPSCs, hESC, and hiPSC-derived HLCs, and PHHs. Red bars represent PHH-unique H3K27ac peaks. (**D**) Nuclear receptor motifs identified as overrepresented binding sites at H3K27ac PHH-unique regions.

The online version of this article includes the following source data and figure supplement(s) for figure 3:

**Source data 1.** Peak annotation results for primary human hepatocyte (PHH)-unique H3K27ac regions.

**Source data 2.** Peak annotation results for hepatocyte-like cell (HLC)-unique H3K27ac regions.

**Source data 3.** Motif enrichment results for primary human hepatocyte (PHH)-unique H3K27ac regions.

**Figure supplement 1.** Comparison of the epigenetic profile of hepatocyte-like cells (HLCs) and primary human hepatocytes (PHHs).

conditions identified above, up to 30 days, and observed the production of polyploid cells with a cobblestone morphology (*Figure 4A*). The hepatocytic identity of these cells was confirmed by the expression of Albumin, *SERPINA1*/A1AT, and AFP in all lines (*Figure 4B, C, D and E*, *Figure 4—figure supplement 1E*). RORc overexpression resulted in a higher number of cells expressing Albumin, which were still heterogeneous across the cell population, potentially representing different subpopulations of HLCs (*Figure 4B*). Moreover, these cells also secreted higher levels of Albumin, although the mRNA expression of this marker was not significantly upregulated (*Figure 4C and D*). In addition, these cells tend to secret lower levels of AFP, although the lower mRNA levels identified were not statistically significant when compared to 3TFs alone (*Figure 4D and E*, *Figure 4—figure supplement 1E*). Notably, CYP3A4 activity levels were significantly higher in cells generated in the presence of RORc, as compared with cells forward programmed with only 3TFs (*Figure 4F*). We next tested whether stimulation with exogenous ligands specific for each nuclear receptor could further induce functional maturation as measured by CYP3A4 activity (desmosterol for RORc, β-estradiol for ERα, and testosterone for AR). Interestingly, only β-estradiol treatment resulted in a three fold increase in CYP3A4 activity, whereas testosterone treatment significantly decreased CYP3A4 activity and desmosterol had no effect (*Figure 4G*). This increase was not observed with 3TFs Fop-Heps thereby suggesting the effect of these ligands was linked to the overexpression of their receptor.

RORc-generated FoP-Heps appeared to have the highest level of functionality and thus, we decided to validate the potential of this combination of factors in an alternative pluripotent stem cell line. We generated Opti-OX hiPSC with the 3TFs or the 3TFs + RORc FoP system, validated the upregulation of these factors after 24 hr of dox treatment (*Figure 4—figure supplement 2A*) and then induced differentiation following the protocol established above. FoP-Heps derived from hiPSC also displayed cobblestone morphology (*Figure 4—figure supplement 2B*) and expressed Albumin, AFP, and *SERPINA1*/A1AT (*Figure 4—figure supplement 2C,D,E,F,G*). Concordantly, a higher number of Albumin positive cells were detected at day 20, as well as higher levels of secreted Albumin and transcript, when RORc was overexpressed (*Figure 4—figure supplement 2C,F,G*). In particular, *ALB* transcript further increased by day 30, suggesting that this cell background might require additional time to achieve maturation. In addition, the presence of RORc significantly increased basal CYP3A4 activity thereby confirming the positive effect of this factor on functional maturity of FoP-Heps (*Figure 4—figure supplement 2H*). Overall, these results showed that overexpression of specific nuclear receptors was compatible with the generation of FoP-hepatocytes. In particular, overexpression of RORc could improve the functionality of hepatocytes generated by overexpression of three LETFs confirming the role of this nuclear receptor in hepatocyte maturation.

## 4TF FoP-Heps display functional characteristics in vitro

Next, we sought to further characterise the functionality of the 4TF (HNF1A, HNF6, FOXA3, and RORc) FoP-Heps derived from either hESC (eFoP-Heps) or hiPSCs (iFoP-Heps) in comparison with HLCs generated by the direct differentiation and PHHs. CYP3A4 activity was significantly higher in 4TF FoP-Heps forward programmed after 20 days than those achieved by HLCs after 30 days of directed differentiation (*Figure 5A*). In addition, we analysed the expression of markers associated with hepatic metabolic functions such as phase I (cytochrome P450 enzymes) and phase II (UGTs) biotransformation, gluconeogenesis (*G6PC* and *PCK1*), and lipid (*PPARα*, *PPARγ*, *FASN*, and *APOA1*) and bile acid (*NR1H4*) metabolism. FoP-Heps expressed a range of these functional markers confirming the acquisition of hepatic functionality (*Figure 5B and C*, *Figure 5—figure supplement 1A*). Overall, the levels of expression achieved by forward programming were equivalent to those achieved by

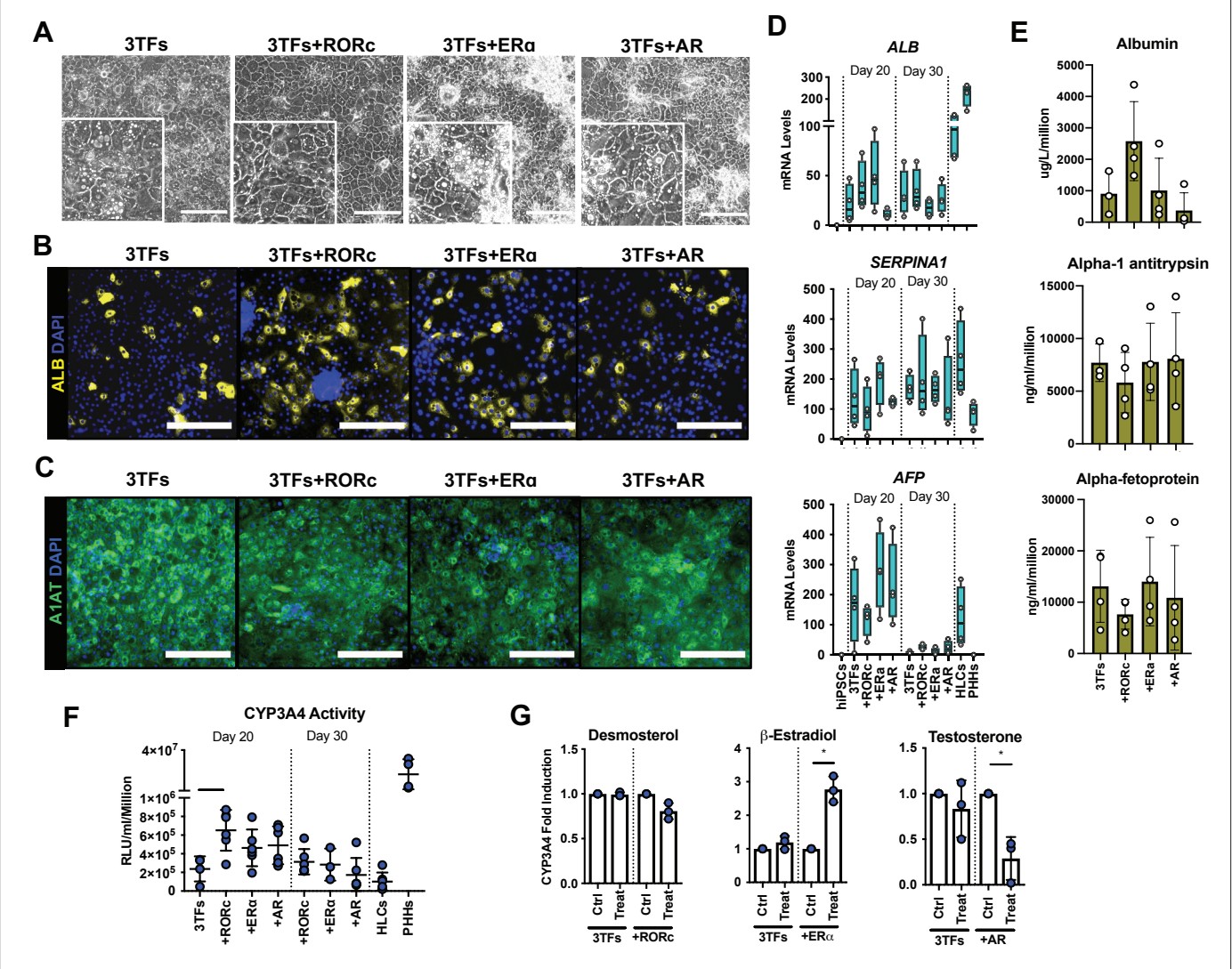

**Figure 4.** Forward programming of human embryonic stem cells (hESCs) into hepatocytes with nuclear receptors. (**A**) Phase contrast images and (**B**) immunofluorescence staining for Albumin (yellow) and (**C**) A1AT (green) in hESCs forward programmed for 20 days with 3TFs alone or in combination with the nuclear receptors RORc, ERα, and AR. Nuclei were counterstained with DAPI (blue). Scale bars, 200 μm. (**D**) mRNA levels of hepatocyte markers (*ALB*, *SERPINA1*, and *AFP*) in FoP-Heps generated with 3TFs alone or in combination with nuclear receptors for 20 and 30 days (n=4). Expression data was normalised to the average of two housekeeping genes (*PBGD* and *RPLP0*). (**E**) Protein secretion levels of Albumin, A1AT, and AFP in hESC-derived FoP-Heps generated with 3TFs alone or in combination with nuclear receptors for 20 days (n=4). Data was normalised per total cell number (millions). (**F**) CYP3A4 activity levels normalised per cell number (millions) in FoP-Heps targeted with 3TFs with or without nuclear receptors, after 20 and 30 days of forward programming (n=3–6). Statistical differences were calculated with one-way ANOVA, corrected for multiple comparisons compared to 3TFs (day 20). (**G**) CYP3A4 fold induction levels in FoP-Heps treated with 100 nM of the ligands as indicated from day 2. Data is normalised to untreated control at day 20 of forward programming (n=3). Statistical differences were calculated with paired t-test. In all plots, bars represent mean with SD, and individual datapoints are shown for all biological replicates. Hepatocyte-like cells (HLCs) generated by direct differentiation and primary human hepatocytes (PHHs) where plotted as controls for CYP3A4 activity and expression data. Significant p-values are shown at each comparison and indicated as *p<0.05, **p<0.01, ***p<0.001, ****p<0.0001.

The online version of this article includes the following source data and figure supplement(s) for figure 4:

**Source data 1.** Individual measurements and statistical tests related to *Figure 4* and supplements.

**Figure supplement 1.** Validation of inducible overexpression (iOX) of combinations of 3TFs and nuclear receptors.

**Figure supplement 2.** Forward programming of human induced pluripotent stem cells (hiPSCs) into hepatocytes with 4TFs.

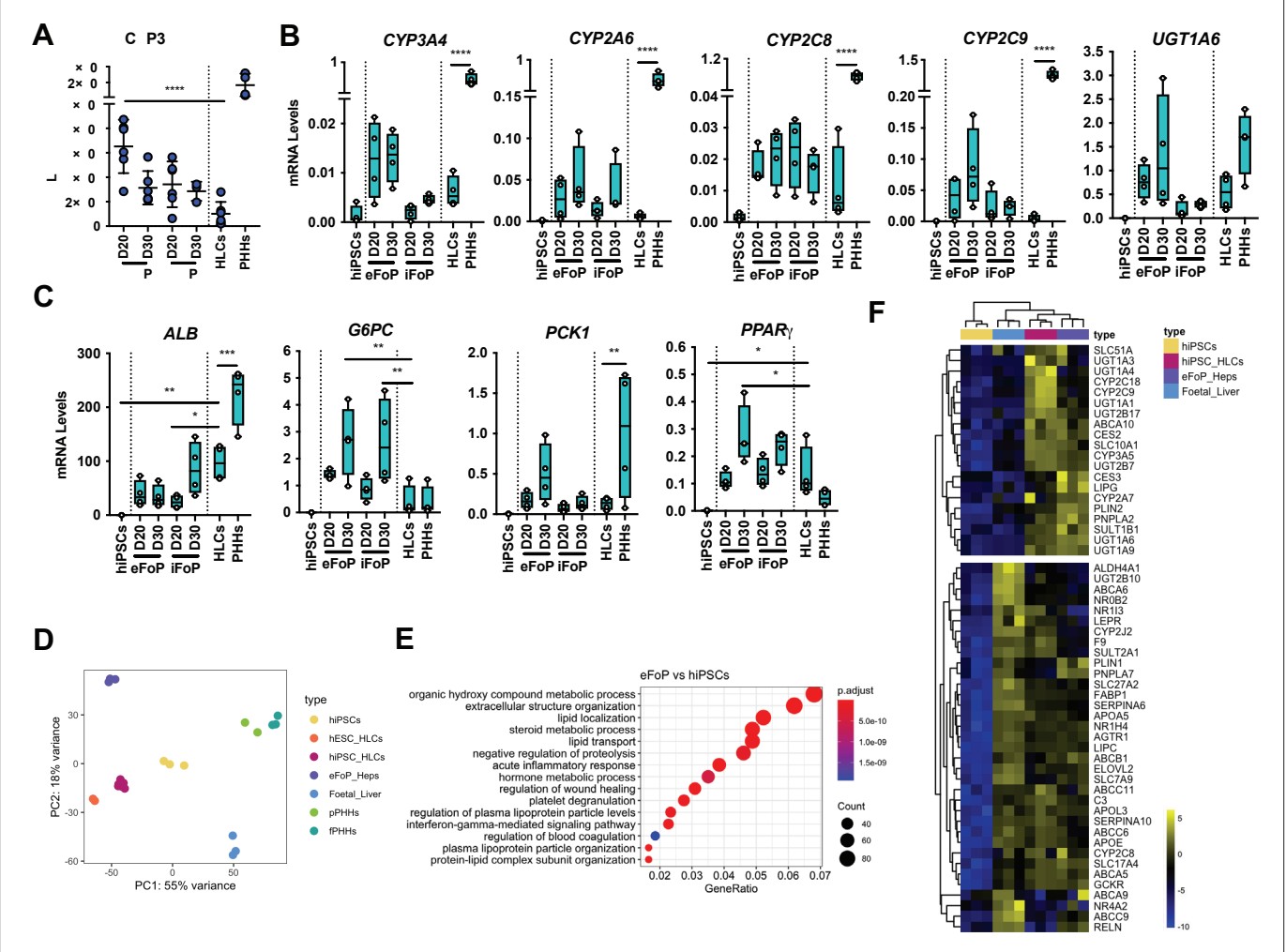

**Figure 5.** 4TF FoP-Heps are transcriptionally equivalent to hepatocyte-like cells (HLCs). (**A**) CYP3A4 activity levels in eFoP-Heps at day 20 (n=6) and day 30 (n=6), iFoP-Heps at day 0 (n=6) and day 30 (n=3), against direct differentiation HLCs (n=6) and primary human hepatocytes (PHHs) (n=4). Statistical differences were tested with one-way ANOVA, corrected for multiple comparisons, between eFoP-Heps group and HLCs. (Statistical test of HLC vs. PHHs can be found in **Figure 2B**) (**B**) mRNA levels of phase I (*CYP3A4*, *CYP2A6,* and *CYP2C8*) and phase II (*UGT1A6*) biotransformation enzymes in 4TF FoP-Heps, HLCs, and PHHs (n=4). (**C**) mRNA level of *ALB*, gluconeogenesis (*G6PC* and *PCK1*), and lipid (*PPARγ*) metabolism in FoP-Heps, HLCs, and PHHs (n=4). Statistical differences were calculated with one-way ANOVA, corrected for multiple comparisons, and all samples compared to HLCs. p-Values are indicated as *$p<0.05$, **$p<0.01$, ***$p<0.001$, ****$p<0.0001$. (**D**) Principal component analysis (PCA) of undifferentiated human induced pluripotent stem cells (hiPSCs), HLCs derived from human embryonic stem cell (hESC) (hESC_HLCs) and hiPSC (hiPSC_HLCs), freshly harvested PHHs (fPHHs) or plated PHHs (pPHHs), foetal livers (FL), and 4TF hESC-derived FoP-Heps (eFoP_Heps). (**E**) Dotplot showing the top 15 hits on gene ontology enrichment analysis on genes associated to genes differentially expressed between eFoP-Heps and undifferentiated hiPSCs. The size of each dot represents number of genes associated to each term and the colours represents the adjusted p-values. (**F**) Heatmap showing expression of genes associated with adult hepatocyte functions found to be expressed in hiPSC-derived HLCs (hiPSC_HLCs) and 4TF hESC-derived FoP-Heps (eFoP_Heps), as compared to undifferentiated hiPSCs. Two clusters were separated as genes expressed (bottom) and not expressed (top) in foetal liver samples (Foetal_Liver).

The online version of this article includes the following source data and figure supplement(s) for figure 5:

**Source data 1.** Individual measurements and statistical tests related to **Figure 5** and corresponding supplements.

**Figure supplement 1.** Characterisation of 4TF FoP-Heps.

**Figure supplement 2.** Characterisation of the transcriptome of 4TF eFoP-Heps.

direct differentiation with the exception of gluconeogenesis genes which were increased in FoP-Heps (**Figure 5C**). Interestingly, expression of gluconeogenesis and lipid metabolism genes was comparable between FoP-Heps and PHHs. However, induction of cytochrome P450 genes remains challenging, indicating that the acquisition of this specific hepatic function could need further refinement of our

protocol (*Figure 5B,*). Of note, in FoP-Heps, the 4TFs remained expressed at physiological levels at the end of our protocol (*Figure 5B*, *Figure 5—figure supplement 1B*). Next, we sought to further characterise the transcriptome of day 20 4TF FoP-Heps by RNA-seq by comparing the transcriptome of eFoP-Heps with undifferentiated hiPSCs, HLCs (derived from direct differentiation of hESC and hiPSCs), adult PHHs, as well as foetal liver cells. As suggested by the marker expression pattern identified by qPCR, the transcriptome of eFoP-Heps closely clustered with HLCs derived from both hESCs and hiPSCs (*Figure 5D*). Interestingly, both eFoP-Heps and HLCs clustered separately from foetal liver cells, indicating that despite these cells not acquiring a fully mature phenotype, these also don't fully resemble a foetal liver stage. We further interrogated the ontology of genes differentially expressed between eFoP and the different groups of samples (*Figure 5—figure supplement 2*). As expected, genes that gain expression in eFoP-Heps compared to undifferentiated hiPSCs are strongly associated with several liver functions such as hormone metabolism, lipid localisation and transport, blood coagulation (*Figure 5E*), further confirming that cells generated by forward programming acquire a hepatocyte identity. Indeed, eFoP-Heps, as well as hiPSC-derived HLCs, expressed a range of adult liver genes that were found to be highly expressed in adult PHHs as identified in *Figure 2*, *Figure 2— source data 2*, including genes that were not expressed in the foetal stage (*Figure 5F*). In addition to expressing mature hepatocyte markers, both eFoP-Heps and iFoP-Heps were also able to uptake low-density lipoprotein (LDL) from the culture medium confirming their capacity to transport lipids (*Figure 6A*). In order to further explore their capacity to metabolise lipids, FoP-Heps were grown in 3D for 5 (D20) or 10 (D30) days as we recently observed that 3D culture conditions facilitate lipid accumulation in HLCs (Carola M Morell, personal communication, *Tilson et al., 2021*). We first confirmed that FoP-Heps grown in 3D retained the expression of hepatocyte markers (*Figure 6B*). Interestingly, *SERPINA1* or *UGT1A6* expression increased in these conditions suggesting an increase in functional maturation promoted by 3D (*Figure 6B*). We then tested the capacity of both eFoP and iFoP-Heps to respond to fatty acids by treating these cells with both oleic acid (OA) and palmitic acid (PA) which are known to induce steatosis and lipotoxicity, respectively (*Ricchi et al., 2009*). In line with their known effect on hepatocytes, OA treatment induced a strong accumulation of lipids as shown by BODIPY staining (*Figure 6C*) while PA treatment induced a reduction in cell viability consistent with lipotoxicity (*Figure 6D*). Thus, FoP-Heps appear to react to fatty acids similarly to their primary counterpart. Finally, we explored the interest of FoP-Heps for modelling the hepatotoxic effect of paracetamol/ acetaminophen (APAP). For that, Fop-Heps were grown in the presence of an APAP dose known to induce liver failure. This treatment resulted in up to 50% reduction in cell viability (*Figure 6E*) suggesting that FoP-Heps could be used for cytotoxic studies. In summary, these results showed that 4TF FoP-Heps derived from either hESC or hiPSCs display characteristics of functional hepatocytes such as expression of genes involved in drug, lipid, glucose and bile acid metabolism, capacity to uptake LDL and fatty acids from the culture medium, as well as response to hepatotoxic factors, demonstrating their potential interest for modelling liver disease in vitro and toxicology screening.

## Discussion

In this study, we have established a method to forward program hPSCs into hepatocytes by taking advantage of the OPTi-OX platform (*Pawlowski et al., 2017*). The success of this approach depends on the selection of TFs, combining factors controlling early liver development and regulators of adult hepatic functions. Nonetheless, most forward programming methods rely on a master regulator to convert hPSCs into a specific cell type. As an example, neurons and muscle cells can be generated by the simple overexpression of NGN2 and MYOD, respectively (*Pawlowski et al., 2017*). Our results show that production of hepatocytes requires a more complex process involving three TFs but also a culture media supporting primary hepatocytes. Furthermore, our best LETFs combination did not include HNF4A, which is known to be a key regulator of hepatocyte function in the adult liver. On the contrary, removing HNF4A significantly improved the identity of the hepatocytes generated. Similar observations were recently reported for the direct reprogramming of human umbilical vein endothelial cells into bipotent hepatocyte progenitor cells where HNF4A was found to be detrimental (*Inada et al., 2020*). HNF4A is essential not only in adult liver but also during development, especially in the establishment of the liver bud (*Gordillo et al., 2015*). Thus, HNF4A might also have a role in preserving foetal liver cells such as hepatoblasts and its overexpression during forward programming

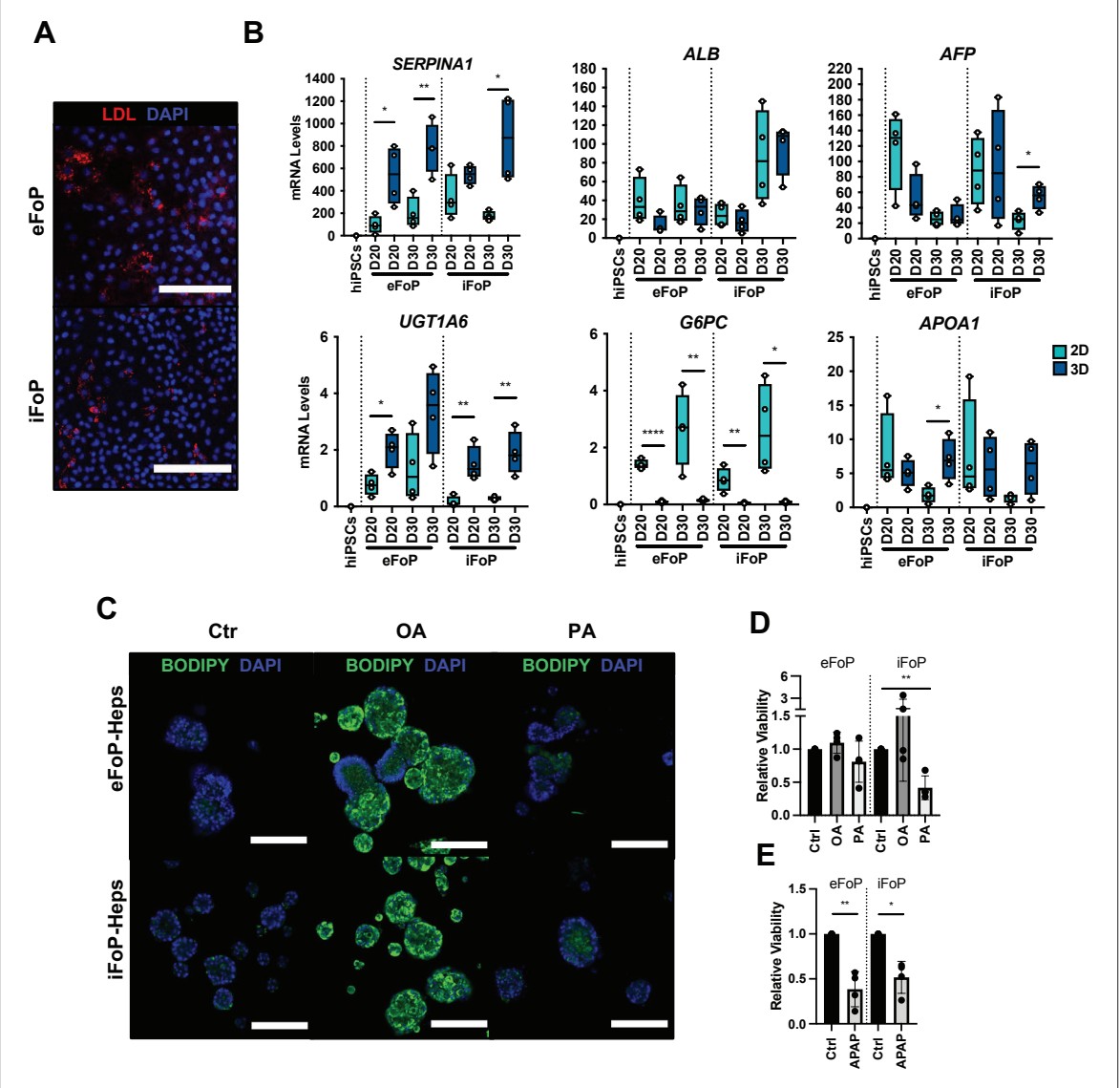

**Figure 6.** RORc promotes functionality of 4TF FoP-Heps. (**A**) Immunofluorescence staining for LDL in FoP-Heps at day 20 of forward programming. Scale bars, 200 μm. Nuclei were counterstained with DAPI (blue). (**B**) Comparison of mRNA levels of *SERPINA1, ALB, AFP, UGT1A6, G6PC,* and *APOA1* in FoP-Heps cultured in 2D and 3D for up to 20 or 30 days of forward programming (n=4). Statistical difference between 2D and 3D were calculated with unpaired t-test. All expression data was normalised to the average of two housekeeping genes (*PBGD* and *RPLP0*). In all plots, bars represent mean with SD, and individual datapoints are shown for all biological replicates. p-Values are indicated as *p<0.05, **p<0.01, ***p<0.001, ****p<0.0001. (**C**) BODIPY staining of FoP-Heps cultured in 3D from day 20 of forward programming and treated with fatty acids (oleic acid [OA], palmitic acid [PA], or BSA [Ctr]) as indicated for 7 days. Scale bars, 200 μm. Nuclei were counterstained with DAPI (blue). (**D**) Cell viability in FoP-Heps treated with the fatty acids as indicated, normalised against FoP-Heps treated with BSA as control (n=4). (**E**) Cell viability in FoP-Heps treated with 25 mM of acetaminophen (APAP) for 48 hr in 3D cultures, normalised against untreated FoP-Heps (n=4). Significant differences were determined with paired t-test. In all plots, bars represent mean with SD, and individual datapoints are shown for all biological replicates. p-Values are indicated as *p<0.05, **p<0.01, ***p<0.001, ****p<0.0001.

The online version of this article includes the following source data for figure 6:

**Source data 1.** Individual measurements and statistical tests related to *Figure 6*.

could block the acquisition of an adult hepatocytic identity. This example illustrates the challenges to identify factors which are uniquely expressed in the adult liver.

Importantly, FoP-Heps generated by LETF overexpression acquired an hepatocytic identity with reduced adult functions, suggesting that this cocktail of TFs might only convert hiPSCs into foetal-like cells. Thus, we decided to add factors which could direct functional maturation of the liver. This latest

category of factors was identified by performing a transcriptomic and epigenetic comparison of PHHs and HLC generated by direct differentiation. The focus on HLCs was based on their well-characterised foetal state and also the broad experience with the cells. These analyses identified a subset of nuclear receptors that were exclusively expressed in the adult liver thereby confirming the relevance of our approach. Of particular interest, RORc, ERα, and AR were identified as key candidates for controlling functional maturation in hepatocytes. Importantly, nuclear receptors are well known to control diverse liver functions including lipid and glucose homeostasis, bile acid clearance, xenobiotic sensing, and regeneration (*Rudraiah et al., 2016*). Both steroid hormonal receptors ERα and AR have been shown to have roles in the regulation of energy homeostasis in the liver (*Shen and Shi, 2021*). Moreover, ERα is involved in cholesterol clearance (*Zhu et al., 2018*) and has also been associated with liver regeneration (*Kao et al., 2018*) and bilirubin metabolism through CYP2A6 (*Kao et al., 2017*). RORc is a nuclear receptor expressed in peripheral tissues including liver, muscle, and adipose tissue and has been proposed to function as an intermediary between the circadian clock and glucose/lipid metabolism (*Cook et al., 2015*). Moreover, *RORγ*-deficient mice exhibit insulin sensitivity and reduced expression of gluconeogenesis, lipid metabolic markers, and a subset of phase I enzymes involved in bile acid synthesis and phase II enzymes (*Kang et al., 2007*; *Takeda et al., 2014a*; *Takeda et al., 2014b*). Based on these previous reports, we propose that the overexpression of RORc and other nuclear receptors could improve specific functions in FoP-Heps by activating a subset of target genes in the hepatic context induced by the LETFs overexpression. Importantly, hepatocyte functionality is spatially different across the liver lobule, being influenced by the gradient of oxygen, nutrients, and signalling (*Trefts et al., 2017*). This hepatic zonation drives different metabolic processes in regards to glucose, lipids, iron, or even xenobiotics, which are under the control of different transcriptomic programs (*Halpern et al., 2017*). Thus, we expect that different combinations of nuclear factors in the background induced by LETFs overexpression could enable the production of hepatocytes with a distinct repertoire of functions.

FoP-Heps generated with the overexpression of the 4TFs (HNF1A, HNF6, FOXA3, and RORc) displayed functional features of adult hepatocytes including Albumin and A1AT secretion, basal CYP3A4 activity, expression of phase I/phase II enzymes, gluconeogenesis and lipid metabolism markers, capacity to uptake LDL and fatty acids, as well as response to toxic compounds. Nonetheless, *CYP3A4* expression remains limited and this gene remains difficult to induce in vitro. Thus, additional TFs could be necessary to generate FoP-Heps exhibiting the full spectrum of functional activities displayed by PHHs. Similarly, culture conditions could be further improved to support key hepatic functions. Indeed, the basal medium used in our protocol does not prevent dedifferentiation of PHHs and thus might not be compatible with the production of fully functional cells by forward programming. Nonetheless, the forward programming method established here presents several advantages over conventional directed differentiation protocols. This is a robust two-step method which bypasses the need for multi-step differentiations which are often associated with batch-to-batch variability. Furthermore, forward programming is faster, generating functional cells in 20 days, as opposed to 30–35 days for direct differentiation. Finally, the yield of cells seems favourable and compatible with large-scale production. Indeed, we observed that forward programming was associated with a six- to eight fold increase in cell number during differentiation while the yield of direct differentiation is lower (data not shown). Moreover, the phenotype achieved is stable even in the absence of dox which allows this method to be applicable in cell therapy and drug discovery (data not shown).

Taken together, our results describe the first method for generating hepatocytes using forward programming. This approach represents the first step towards the high-throughput and large-scale production of specialised hepatocytes displaying a spectrum of functions relevant for different applications in disease modelling and drug screening.

## Materials and methods

### hPSC culture

The human ESC H9 (WiCell) and IPSC A1ATD[R/R] (*Yusa et al., 2011*) lines were used in this project. Human iPSC line was derived as previously described, under approval by the regional research ethics committee (REC 08/H0311/201). Both hPSCs were cultured on vitronectin XFTM (10 µg/ml, StemCell

Technologies)-coated plates and in Essential 8 (E8) chemically defined medium consisting of DMEM/ F12 (Gibco), L-ascorbic acid 2-phosphate (1%), insulin-transferrin-selenium solution (2%, Life Technologies), sodium bicarbonate (0.7%), and Penicillin/Streptomycin (1%), freshly supplemented with TGFβ (10 ng/ml, R&D) and FGF2 (12 ng/ml, Qkine) (*Chen et al., 2011*). For routine dissociation, cells were incubated with 0.5 μM EDTA (Thermo Fisher Scientific) for 3 min at 37°C seeded in small clumps. Cells were maintained at 37°C in 20% $O_2$, 5% $CO_2$ and medium was replenished every 24 hr.

Authentication of hPSCs was achieved by confirming the expression of pluripotency genes. Cells were routinely confirmed to be mycoplasma free using broth and PCR-based assays. The cell lines are not on the list of commonly misidentified cell lines (International Cell Line Authentication Committee).

## Gene targeting

Inducible hESC and hiPSC lines were generated using the OPTi-OX system as previously described (*Bertero et al., 2016*; *Pawlowski et al., 2017*). Briefly, two gene safe harbours were targeted (GSH).

**Table 1.** Sequences of primers used for cloning.

| Gene | Primer | Sequence 5'–3' (GSG linker sequence underlined) |
|---|---|---|
| HNF1A | Start_F | CAC TTT TGT CTT ATA CTT ACT AGT GCC ACC ATG GTT TCT AAA CTG AGC CAG CTG CAG |
| HNF1A | P2A_R | TTC CAC GTC TCC TGC TTG CTT TAA CAG AGA GAA GTT CGT GGC <u>TCC GGA GCC</u> CTG GGA GGA AGA GGC CAT CTG G |
| HNF4A | E2A_F | TAT GCT CTC TTG AAA TTG GCT GGA GAT GTT GAG AGC AAC CCT GGA CCT GTC AGC GTG AAC GCG CCC CT |
| HNF4A | Stop_R | AGA GGA TCC CCG GGT ACC GAG CTC GAA TTC CTA GAT AAC TTC CTG CTT GGT GAT GGT CG |
| HNF4A | P2A_F | TCT CTG TTA AAG CAA GCA GGA GAC GTG GAA GAA AAC CCC GGT CCT GTC AGC GTG AAC GCG CCC CT |
| HNF4A | T2A_R | CTC CTC CAC GTC ACC GCA TGT TAG AAG ACT TCC TCT GCC CTC <u>TCC GGA GCC</u> GAT AAC TTC CTG CTT GGT GAT GGT CG |
| HNF4A | Start_F | CAC TTT TGT CTT ATA CTT ACT AGT GCC ACC ATG GTC AGC GTG AAC GCG CCC |
| HNF4A | P2A_R | TTC CAC GTC TCC TGC TTG CTT TAA CAG AGA GAA GTT CGT GGC <u>TCC GGA GCC</u> GAT AAC TTC CTG CTT GGT GAT GGT CG |
| FOXA3 | T2A_F | AGT CTT CTA ACA TGC GGT GAC GTG GAG GAG AAT CCC GGC CCT CTG GGC TCA GTG AAG ATG GAG GC |
| FOXA3 | E2A_R | CTC AAC ATC TCC AGC CAA TTT CAA GAG AGC ATA ATT AGT ACA CTG <u>TCC GGA GCC</u> GGA TGC ATT AAG CAA AGA GCG GGA ATA G |
| FOXA3 | P2A_F | TCT CTG TTA AAG CAA GCA GGA GAC GTG GAA GAA AAC CCC GGT CCT CTG GGC TCA GTG AAG ATG GAG GC |
| FOXA3 | T2A_R | CTC CTC CAC GTC ACC GCA TGT TAG AAG ACT TCC TCT GCC CTC <u>TCC GGA GCC</u> GGA TGC ATT AAG CAA AGA GCG GGA ATA G |
| FOXA3 | T2A_F | AGT CTT CTA ACA TGC GGT GAC GTG GAG GAG AAT CCC GGC CCT CTG GGC TCA GTG AAG ATG GAG GC |
| FOXA3 | Stop_R | AGA GGA TCC CCG GGT ACC GAG CTC GAA TTC CTA GGA TGC ATT AAG CAA AGA GCG GGA ATA G |
| HNF6 | P2A_F | TCT CTG TTA AAG CAA GCA GGA GAC GTG GAA GAA AAC CCC GGT CCT AAC GCG CAG CTG ACC ATG GAA GC |
| HNF6 | T2A_R | CTC CTC CAC GTC ACC GCA TGT TAG AAG ACT TCC TCT GCC CTC <u>TCC GGA GCC</u> TGC TTT GGT ACA AGT GCT TGA TGA AGA GAT |
| HNF6 | T2A_F | AGT CTT CTA ACA TGC GGT GAC GTG GAG GAG AAT CCC GGC CCT AAC GCG CAG CTG ACC ATG GAA GC |
| HNF6 | Stop_R | AGA GGA TCC CCG GGT ACC GAG CTC GAA TTC CTA TGC TTT GGT ACA AGT GCT TGA TGA AGA GAT |
| RORy | E2A_F | TAT GCT CTC TTG AAA TTG GCT GGA GAT GTT GAG AGC AAC CCT GGA CCT GAC AGG GCC CCA CAG AGA CAG |
| RORy | Stop_R | AGA GGA TCC CCG GGT ACC GAG CTC GAA TTC CTA CTT GGA CAG CCC CAC AGG TGA C |
| ESR1 | E2A_F | TAT GCT CTC TTG AAA TTG GCT GGA GAT GTT GAG AGC AAC CCT GGA CCT ACC ATG ACC CTC CAC ACC AAA GCA |
| ESR1 | Stop_R | AGA GGA TCC CCG GGT ACC GAG CTC GAA TTC CTA GAC CGT GGC AGG GAA ACC CTC |
| AR | E2A_F | TAT GCT CTC TTG AAA TTG GCT GGA GAT GTT GAG AGC AAC CCT GGA CCT GAA GTG CAG TTA GGG CTG GGA AG |
| AR | Stop_R | AGA GGA TCC CCG GGT ACC GAG CTC GAA TTC CTA CTG GGT GTG GAA ATA GAT GGG CTT G |

The hROSA26 locus was targeted with a constitutively expressed transactivator (rtTA) and the AAVS1 locus with the transgenes of interest under a TET-responsive element (TRE). Different combinations of TFs and/or nuclear receptors as stated throughout the manuscript were cloned. Template cDNA sequences were obtained either from Dharmacon: *HNF6* (MHS6278-213244170), *HNF1A* (MHS6278-202857902), *RORγ* (MHS6278-202800991), and *ESR1* (MHS6278- 211691051); or amplified from human primary liver cDNA: *HNF4A*, *FOXA3*, and *AR*. Sequences were amplified using the KAPA HiFi HotStart ReadyMix (Roche). The primers used to amplify and clone the sequences into the backbone vector contained upstream and downstream overhangs in order to generate a GSG (Gly-Ser-Gly) linker and a different 2A peptide as listed in *Table 1*. The different vectors were constructed by Gibson Assembly (New England Biolabs) using a 1:3 pmol ratio of vector to insert. For targeting, hPSCs were dissociated into single cells with StemPro accutase (Thermo Fisher) for 5 min, and 1 million cells were transfected with 2 µg of donor vector and 2 µg of each AAVS1 ZFN expression plasmids using the P3 Primary Cell 4D-Nucleofector X Kit (Lonza). Cells were seeded in E8 medium supplemented with 10 µM ROCK Inhibitor Y-27632 (Selleckchem). After 5–7 days, colonies were selected with 1 µg/ml puromycin (Sigma-Aldrich) for at least 2 days, after which they were individually picked and genotyped as previously described (*Bertero et al., 2016*; *Pawlowski et al., 2017*).

## Hepatocyte direct differentiation

hPSCs were dissociated into single cells following incubation with StemPro Accutase (Thermo Fisher) for 5 min at 37°C and seeded at a density of 50,000 cells/cm$^2$ in E8 medium supplemented with 10 µM ROCK Inhibitor Y-27632 (Selleckchem). Hepatocytes were differentiated 48 hr after seeding, as previously reported (*Hannan et al., 2013*) with minor modifications. Following endoderm differentiation, anterior foregut specification was achieved with RPMI-B27 differentiation media supplemented with 50 ng/ml Activin A (R&D) for 5 days. Cells at the foregut stage were further differentiated into hepatocytes with Hepatozyme complete medium: HepatoZYME-SFM (Thermo Fisher) supplemented with 2 mM L-glutamine (Thermo Fisher), 1% penicillin-streptomycin (Thermo Fisher), 2% non-essential amino acids (Thermo Fisher), 2% chemically defined lipids (Thermo Fisher), 14 µg/ml of insulin (Roche), 30 µg/ml of transferrin (Roche), 50 ng/ml hepatocyte growth factor (R&D), and 20 ng/ml oncostatin M (R&D), for up to 27 days.

## Forward programming into hepatocytes

hPSCs were dissociated into single cells following incubation with StemPro Accutase (Thermo Fisher) for 5 min at 37°C and seeded at a density of 40–50,000 cells/cm$^2$ in E8 medium supplemented with 10 µM ROCK Inhibitor Y-27632 (Selleckchem). E8 medium was replenished the following day. Following 48 hr, initial induction of the transgenes was achieved by incubation in E6 medium (E8 without growth factors) supplemented with 1 mg/ml dox for 24 hr. Cells were then maintained in Hepatozyme complete medium supplemented with 1 mg/ml dox for the remaining duration of the protocol. Medium was replenished every day for the next 4 days, and every other day here after. For specific experiments, cell lines were treated with 100 nM of desmosterol, testosterone, or β-estradiol (E2) from day 2 of forward programming. All ligands were purchased from Sigma-Aldrich and reconstituted in ethanol. For 3D cultures, forward programmed cells were embedded in Matrigel Growth Factor Reduced Basement Membrane Matrix, Phenol Red-free (Corning) at day 15 or day 20 and cultured for 5 or 10 days, respectively. Cells were dissociated with Hank's based cell dissociation buffer (Gibco) for 20 min at 37°C, resuspended in Matrigel and seeded in 40–50 µl domes in Hepatozyme complete medium supplemented with 1 mg/ml dox.

## Primary liver samples

Fresh primary hepatocytes used for RNA-seq were obtained as previously reported (*Segeritz et al., 2018*). Primary plated hepatocytes from four donors (three males and one female) were purchased from Biopredic International (Rennes, France), meeting the manufacturer's quality control requirements. Cells were maintained in short-term monolayer cultures in William's E (Gibco) supplemented with 1% glutamine (Gibco), 1% penicillin-streptomycin (Gibco), 700 nM insulin (Sigma-Aldrich), and 50 µM hydrocortisone (Sigma). Functional assays such as CYP3A4 activity measurement were performed in Hepatozyme complete medium within 48 hr of receipt. Bulk foetal tissue was obtained from patients

undergoing elective terminations up to the third trimester, under approval by the regional research ethics committee (REC- 96/085). The tissue was lysed and RNA harvested as indicated below.

## CYP3A4 assay

Measurement of CYP3A4 enzymatic activity was performed using the P450 Glo kit (Promega). Cells were incubated with 1:1000 luciferin-IPA in Hepatozyme complete for 1 hr at 37°C. Supernatant was mixed with detection reagent in a 1:1 ratio and incubated at room temperature for 20 min in Greiner white 96-well microplates (Sigma-Aldrich). Luminescence was measured in triplicate on a GloMax plate reader. Hepatozyme complete medium was used as background control. Relative light units were normalised for background, volume and average total number of cells obtained after differentiation.

## LDL uptake assay

LDL uptake capacity was measured with the LDL Uptake Assay Kit (Abcam). Cells were incubated with 1:100 human LDL conjugated to DyLight 550 in Hepatozyme complete medium for 3 hr at 37°C. Cells were then washed and fixed with 4% PFA for 20 min at 4°C.

## Fatty acid treatments

Forward programmed cells were embedded in 3D from day 20 and cultured for 7 days in Hepatozyme complete medium supplemented with either BSA (control), or OA (0.25 mM) or PA (0.25 mM) conjugated with BSA. Intracellular lipid accumulation was detected by incubating cells with 1 µl/ml Bodipy (Thermo Fisher) for 30 min, followed by DAPI (Hoechst) diluted 1:10,000 in PBS for 30 min and imaged on a Zeiss LSM 700 confocal microscope.

## APAP toxicity

The hepatotoxicity of APAP was tested by incubating forward programmed cells cultured in 3D from day 15 in Hepatozyme complete medium supplemented with 25 mM APAP (R&D) for 48 hr (day 18 to day 20) after which cell viability was determined.

## Cell viability

Cell viability was determined by incubating cells with 1:10 Presto Blue reagent (Invitrogen) in Hepatozyme complete medium at 37°C for 4 hr. Fluorescence was measured using the EnVision plate reader with an excitation emission of 560 nm/590 nm.

## RT-qPCR

RNA was extracted from either cells or tissues using GenElute Mammalian Total RNA Miniprep Kit (Sigma-Aldrich) according to the manufacturer's instructions; 500 ng of RNA were reverse transcribed into cDNA using Random Primers and SuperScript II (Invitrogen) according to the manufacturer's instructions. qPCR was performed using the KAPA SYBR FAST qPCR Kit low-ROX (Sigma-Aldrich) with 200 nM of forward and reverse primers (Sigma-Aldrich; primers listed in *Table 2*) on a QuantStudio 5 (Applied Biosystems). qPCRs were performed in technical duplicates and normalised to the average of two housekeeping genes (*RPLP0* and *PBGD*) using the $2^{-\Delta Ct}$ method.

## Immunofluorescence staining

Cells in monolayer were fixed in 4% PFA for 20 min at 4°C and blocked for 30 min in 10% donkey serum (Bio-Rad) and 0.1% Triton X-100 (Sigma-Aldrich). Fixed cells were incubated with primary antibodies listed in *Table 3* in 1% donkey serum and 0,01% Triton X-100 overnight at 4 °C. Following washing, cells were incubated with Alexa Fluor 488-, 568-, or 647-conjugated secondary antibodies (Life Technologies) for 1 hr at room temperature diluted in 1% donkey serum and 0.01% Triton X-100. For nuclei visualisation, cells were incubated with adding DAPI/Hoechst 33258 (bis-Benzimide H, Sigma-Aldrich) diluted 1:10,000 in PBS for 10 min at room temperature. Cells were imaged either on a Zeiss Axiovert 200M or on a Zeiss LSM 700 confocal microscope.

## Secreted protein quantification

Albumin, Alpha-Fetoprotein, and Alpha-1-Antitrypsin were measured in the cell culture supernatant of monolayer cultures, which were replenished with fresh Hepatozyme complete medium 24 hr prior to

**Table 2.** Sequences of primers used for qPCR.

| Gene | Forward | Reverse |
| --- | --- | --- |
| AFP | TGCGGCCTCTTCCAGAAACT | TAATGTCAGCCGCTCCCTCG |
| ALB | CCTTTGGCACAATGAAGTGGGTAACC | CAGCAGTCAGCCATTTCACCATAG |
| APOA1 | AGACAGCGGCAGAGACTATG | CCAGTTGTCAAGGAGCTTTAGG |
| CYP2A6 | CAGCACTTCCTGAATGAG | AGGTGACTGGGAGGACTTGAGGC |
| CYP2C8 | CATTACTGACTTCCGTGCTACAT | CTCCTGCACAAATTCGTTTTCC |
| CYP2C9 | GCCGGCATGGAGCTGTTTTTAT | GCCAGGCCATCTGCTCTTCTT |
| CYP3A4 | TGTGCCTGAGAACACCAGAG | GTGGTGGAAATAGTCCCGTG |
| FASN | GCAAGCTGAAGGACCTGTCT | AATCTGGGTTGATGCCTCCG |
| FOXA3 | TGGGCTCAGTGAAGATGGAG | GGGGATAGGGAGAGCTTAGAG |
| G6PC | GTGTCCGTGATCGCAGACC | GACGAGGTTGAGCCAGTCTC |
| HHEX | GCCCTTTTACATCGAGGACA | AGGGCGAACATTGAGAGCTA |
| HNF1A | TGGCCATGGACACGTACAG | GCTGCTTGAGGGTACTTCTG |
| HNF4A | CATGGCCAAGATTGACAACCT | TTCCCATATGTTCCTGCATCAG |
| HNF6 | GTGTTGCCTCTATCCTTCCCAT | CGCTCCGCTTAGCAGCAT |
| NANOG | CATGAGTGTGGATCCAGCTTG | CCTGAATAAGCAGATCCATGG |
| NR1H4 | ACTGAACTCACCCCAGATCAA | TGGTTGCCATTTCCGTCAAA |
| PBGD | GGAGCCATGTCTGGTAACGG | CCACGCGAATCACTCTCATCT |
| PCK1 | ACACAGTGCCCATCCCCAAA | GGTGCGACCTTTCATGCACC |
| POU5F1 | AGTGAGAGGCAACCTGGAGA | ACACTCGGACCACATCCTTC |
| PPARa | CCCTCCTCGGTGACTTATCC | CGGTCGCACTTGTCATACAC |
| PPARy | GAGCCTGCATCTCCACCTTAT | AGAAACCCTTGCATCCTTCACA |
| RORy | CTACGGCAGCCCCAGTTT | GCTGGCATGTCTCCCTGTA |
| RPLP0 | GGCGTCCTCGTGGAAGTGAC | GCCTTGCGCATCATGGTGTT |
| SERPINA1 | CCACCGCCATCTTCTTCCTGCCTGA | GAGCTTCAGGGGTGCCTCCTCTG |
| SOX17 | CGCACGGAATTTGAACAGTA | GGA TCAGGGACCTGTCACAC |
| TBX3 | TGGAGCCCGAAGAAGAGGTG | TTCGCCTTCCCGACTTGGTA |
| UGT1A1 | TGATCCCAGTGGATGGCAGC | CAACGAGGCGTCAGGTGCTA |
| UGT1A6 | GGAGCCCTGTGATTTGGAGAGT | GACCCCGGTCACTGAGAACC |

collection. Concentrations were detected by ELISA (performed by core biomedical assay laboratory, Cambridge University Hospitals) and normalised to cell number.

## RNA-seq analyses

RNA-seq datasets were generated for undifferentiated hiPSCs (n=3), hESC-derived HLCs (n=2), hiPSC-derived HLCs (n=6), fPHHs (n=3), commercially purchased PHHs (pPHHs, n=2), bulk foetal liver samples (FL, n=3), and hESC-derived 4TF FoP-Heps (eFoP, n=3). RNA was extracted from either cells or tissues using GenElute Mammalian Total RNA Miniprep Kit (Sigma-Aldrich) according to the manufacturer's instruction. Poly-A library preparation and sequencing were performed by Cambridge Genomic Services (hESC_HLCs; pPHHs), the Wellcome Trust Sanger Institute (hiPSCs, hiPSC_HLCs, fPHHs), and the Cambridge Stem Cell Institute and CRUK (FL, eFoP). Quality of reads was assessed with FastQC. For consistency, fastq reads were split into single-end reads and trimmed to the same length (40 bp) using cutadapt version 2.10. Single-end fastq files were mapped and quantified using salmon version 1.2.1 with the following parameters: -l A, -GCbias, -posbias, -validatemappings (**Patro**

**Table 3.** Lisf of primary antibodies.

| Protein | Supplier | Catalog number | Host | Concentration |
| --- | --- | --- | --- | --- |
| Albumin | Bethyl Laboratories | A80-229A | goat | 1:100 |
| Alpha-1 Antitrypsin | Dako | A0012 | rabbit | 1:100 |
| Alpha-Fetoprotein | Dako | A0008 | rabbit | 1:100 |
| HNF4A | Abcam | ab92378 | rabbit | 1:100 |
| HNF1A | Santa cruz | sc-135939 | mouse | 1:50 |
| HNF6 | Santa cruz | sc-13050 | rabbit | 1:100 |
| FOXA3 | Santa cruz | sc-166703 | mouse | 1:50 |
| RORc | Abcam | ab221359 | rabbit | 1:100 |
| ERα | Abcam | ab32063 | rabbit | 1:100 |
| AR | Abcam | ab108341 | rabbit | 1:100 |

*et al., 2017*). The index used was pre-built from the human GRCh38 cDNA reference sequence from Ensembl (refgenomes.databio.org). DGE was calculated using DESeq2 (*Love et al., 2014*), with the following parameters: padj > 0.05, basemean > 100 and log2 fold change >2 or <–2 between groups as depicted in each figure. GO enrichment was calculated with the clusterProfiler package (*Yu et al., 2012*). Pathway analysis on significantly misregulated TFs was assessed using ReactomePA (*Yu and He, 2016*). Mouse liver poly-A plus RNA-seq was downloaded from ENCODE (*ENCODE Project Consortium, 2012*). Single-end fastq reads were trimmed in both replicates from each dataset to 70 bp using cutadapt version 2.10. Fastq were mapped and quantified using salmon version 1.2.1 with the following parameters: -l A, -Gbias, -seqbias, -validatemappings using a pre-build mm10 cDNA reference genome. DeSeq2 was used to generate all plots for visualisation.

## Chromatin immunoprecipitation

ChIP was performed as previously reported (*Brown et al., 2011*). Briefly, chromatin was crosslinked with 1% formaldehyde (Sigma-Aldrich) for 10 min at room temperature and quenched with 0.125 M glycine (Sigma-Aldrich). Cells and nuclei were subsequently lysed and chromatin was sonicated to fragment DNA to about 200–500 bp on a Bioruptor Pico sonication device (Diagenode). Sonicated chromatin was pre-cleared with same-host IgG and protein G Dynabeads (Thermo Fisher), 100 μg of cleared chromatin (protein) was incubated with 2 μg of the following antibodies overnight at 4°C: H3K27ac (Abcam, ab4729), H3K4me1 (Abcam, ab8895), H3K27me3 (Active Motif, 39155), and H3K4me3 (Merk, 05-745R), after which complexes captured with 30 μl of protein G Dynabeads (Thermo Fisher). Complexes were washed, RNAse A (Thermo Fisher) and Proteinase K (Sigma-Aldrich) treated, and DNA was purified by phenol-chloroform extraction and precipitated with GlycoBlue (Thermo Fisher), sodium acetate (Thermo Fisher), and ethanol (Sigma-Aldrich). A sonicated chromatin sample (1%) was also collected as input for normalisation and 10 ng of DNA were used for ChIP-seq library preparations.

## ChIP-seq analyses

Library preparation and sequencing and alignment were performed by the Wellcome Trust Sanger Institute DNA Sequencing Facility (Hinxton, UK). Sequencing was performed on an Illumina HiSeq v4 to obtain paired-end reads with 75 bp length. ChIP-seq reads were mapped to human genome assembly GRCh38 with bwa. Aligned data in BAM format was sorted and indexed with samtools. Coverage files were generated using deeptools bamCoverage, with a bin size of 10 bp and normalised as RPKM for visualisation in IGV and heatmap representation with deeptools. In order to plot PCA, average scores were calculated over 1000 bp bins. For peak calling, BAM files were converted to SAM and peaks called using homer (*Heinz et al., 2010*). Both replicates were used for peak calling against input with disabled local filtering invoking the following flags for H3K27ac: -region -L 0. In order to identify regulatory regions specifically active in PHH or HLCs, differentially bound peaks were determined using PHH datasets as target against all HLCs datasets as background, and vice

versa, with a fold enrichment over background of 4. A list of genes annotated for each peak dataset can be found in, *Figure 3—source data 1 and 2*. For motif enrichment, peak calling was performed on nucleosome-free regions by invoking the flags -L 1 -nfr, in order to determine the 'dips' within H3K27ac-rich regions. These sets of regions were overlapped with the differentially bound peaks as above, in order to perform PHH or HLC-specific motif enrichment. Peak annotation and GO enrichment were determined with the clusterProfiler R package (*Yu et al., 2012*). Undifferentiated hiPSCs ChIP-seq reads aligned to the same genome assembly were downloaded from ENCODE (*ENCODE Project Consortium, 2012*) and treated as above.

## Statistical analysis

Statistical analyses were conducted using GraphPad 9.0.0 and specific tests are indicated in the figure legends. For each figure, sample size n indicates the number of independent experiments or biological replicates and individual values are represented for every graph. Testing between groups was performed with at least n≥3 independent experiments and p-value groups are indicated within the figure where significant.

## Acknowledgements

This work was supported by funding from the European Research Council Grant New-Chol, the UK Regenerative Medicine Platform, and a core support grant from the Wellcome MRC – Cambridge Stem Cell Institute. We thank Anna Osnato and Pedro Madrigal for bioinformatics support, and Stephanie Brown for technical advice. We acknowledge the Wellcome Trust Sanger Institute sequencing platform, the ENCODE Consortium, and the ENCODE production laboratories in generating the particular datasets used in this manuscript.

## Additional information

### Competing interests

Fabian Bachinger: is a PhD student sponsored by bit.bio. Ludovic Vallier: is a founder and shareholder of DefiniGEN, Aculive Therapeutics and Billitech. The other authors declare that no competing interests exist.

### Funding

| Funder | Grant reference number | Author |
| --- | --- | --- |
| European Research Council | New-Chol | Rute A Tomaz<br>Ludovic Vallier |
| UK Regenerative Medicine Platform | | Rute A Tomaz<br>Rute A Tomaz |
| Wellcome - MRC Cambridge Stem Cell Institute, University of Cambridge | | Annabelle Wurmser<br>Annabelle Wurmser<br>Annabelle Wurmser |
| Gates Cambridge Trust | PhD Studentship | Brandon T Wesley |
| Chan Zuckerberg Initiative | | Carola M Morell<br>Carola M Morell |
| bit.bio | PhD Studentship | Fabian Bachinger |

The funders had no role in study design, data collection and interpretation, or the decision to submit the work for publication.

### Author contributions

Rute A Tomaz, Conceptualization, Software, Formal analysis, Validation, Investigation, Methodology, Writing – original draft; Ekaterini D Zacharis, Fabian Bachinger, Annabelle Wurmser, Dominika Dziedzicka, Validation, Investigation; Daniel Yamamoto, Software, Formal analysis; Sandra

Petrus-Reurer, Investigation; Carola M Morell, Brandon T Wesley, Resources; Imbisaat Geti, Resources, Generation of RNA-seq dataset; Charis-Patricia Segeritz, Resources, Generation of RNA-seq dataset; Miguel C de Brito, Resources, Generation of RNA-seq dataset; Mariya Chhatriwala, Resources, Generation of RNA-seq dataset; Daniel Ortmann, Methodology; Kourosh Saeb-Parsy, Supervision; Ludovic Vallier, Conceptualization, Supervision, Funding acquisition, Project administration, Writing – review and editing

### Author ORCIDs
Rute A Tomaz (ID) http://orcid.org/0000-0002-9377-1431
Ludovic Vallier (ID) http://orcid.org/0000-0002-3848-2602

### Decision letter and Author response
Decision letter https://doi.org/10.7554/eLife.71591.sa1
Author response https://doi.org/10.7554/eLife.71591.sa2

## Additional files

### Supplementary files
• Transparent reporting form

### Data availability
RNA-seq datasets used in this study are accessible on Array Express under the accession number E-MTAB-10634 and E-MTAB-11852. In addition, 3 of the hiPSC_HLCs data sets have been previously deposited with the accession number E-MTAB-6781 (Segeritz et al., 2018). *Mus musculus* C57BL/6 liver embryo RNA-seq datasets were obtained from the ENCODE database (Nakamori et al., 2016) (https://www.encodeproject.org/) with the following accession numbers: ENCSR216KLZ (E12.5 liver), ENCSR826HIQ (E16.5 liver), ENCSR096STK (P0 liver), ENCSR000BYS (8 weeks mixed sex adult liver) and ENCSR216KLZ (10 weeks adult liver). ChIP-seq datasets generated in this study have been deposited on Array Express with the accession number E-MTAB-10637, and publicly available datasets for hiPSCs were used from the ENCODE database with the following accession numbers: ENCSR729ENO (H3K27ac), ENCSR249YGG (H3K4me1), ENCSR386RIJ (H3K27me3), ENCSR657DYL (H3K4me3) and ENCSR773IYZ (input). All data generated or analysed during this study is included in the manuscript, supporting files, and source data files.

The following datasets were generated:

| Author(s) | Year | Dataset title | Dataset URL | Database and Identifier |
|---|---|---|---|---|
| Tomaz RA, Zacharis ED, Bachinger F, Wurmser A, Yamamoto D, Petrus-Reurer S, Morell CM, Dziedzicka D, Wesley BT, Geti I, Segeritz CP, de Brito MC, Chhatriwala M, Ortmann D, Saeb-Parsy K, Vallier L | 2022 | RNA-sequencing of forward programmed hepatocytes and fetal liver samples | https://www.ebi.ac.uk/arrayexpress/experiments/E-MTAB-11852 | ArrayExpress, E-MTAB-11852 |
| Tomaz RA, Zacharis ED, Bachinger F, Wurmser A, Yamamoto D, Petrus-Reurer S, Morell CM, Dziedzicka D, Wesley BT, Geti I, Segeritz CP, de Brito MC, Chhatriwala M, Ortmann D, Saeb-Parsy K, Vallier L | 2022 | ChIP-sequencing of hepatocyte-like cells (HLCs) and primary human hepatocytes (PHHs) | https://www.ebi.ac.uk/arrayexpress/experiments/E-MTAB-10637 | ArrayExpress, E-MTAB-10637 |

*Continued on next page*

*Continued*

| Author(s) | Year | Dataset title | Dataset URL | Database and Identifier |
|---|---|---|---|---|
| Tomaz RA, Zacharis ED, Bachinger F, Wurmser A, Yamamoto D, Petrus-Reurer S, Morell CM, Dziedzicka D, Wesley BT, Geti I, Segeritz CP, de Brito MC, Chhatriwala M, Ortmann D, Saeb-Parsy K, Vallier L | 2022 | RNA-sequencing of hepatocyte-like cells (HLCs) and primary human hepatocytes (PHHs) | https://www.ebi.ac.uk/arrayexpress/experiments/E-MTAB-10634 | ArrayExpress, E-MTAB-10634 |

The following previously published datasets were used:

| Author(s) | Year | Dataset title | Dataset URL | Database and Identifier |
|---|---|---|---|---|
| Segeritz CP, Rashid ST, de Brito MC, Serra MP, Ordonez A, Morell CM, Kaserman JE, Madrigal P, Hannan NRF, Gatto L, Tan L, Wilson AA, Lilley K, Marciniak SJ, Gooptu B, Lomas DA, Vallier L | 2018 | RNA-seq of hepatocytes obtained through step-wise differentiation of hIPSCs from a patient with A1AT deficiency and its point mutation-corrected isogenic hIPSC line. Comparison to primary hepatocytes from a healthy donor and an A1AT-deficient patient | https://www.ebi.ac.uk/arrayexpress/experiments/E-MTAB-6781 | ArrayExpress, E-MTAB-6781 |
| Encode consortium | 2015 | polyA plus RNA-seq | https://doi.org/10.17989/ENCSR216KLZ | Encode portal, 10.17989/ENCSR216KLZ |
| Encode consortium | 2016 | Histone ChIP-seq | https://doi.org/10.17989/ENCSR729ENO | Encode portal, 10.17989/ENCSR729ENO |
| Encode consortium | 2015 | polyA plus RNA-seq | https://doi.org/10.17989/ENCSR826HIQ | Encode portal, 10.17989/ENCSR826HIQ |
| Encode consortium | 2015 | polyA plus RNA-seq | https://doi.org/10.17989/ENCSR096STK | Encode portal, 10.17989/ENCSR096STK |
| Encode consortium | 2015 | polyA plus RNA-seq | https://doi.org/10.17989/ENCSR000BYS | Encode portal, 10.17989/ENCSR000BYS |
| Encode consortium | 2016 | Histone ChIP-seq | https://doi.org/10.17989/ENCSR249YGG | Encode portal, 10.17989/ENCSR249YGG |
| Encode consortium | 2016 | Histone ChIP-seq | https://doi.org/10.17989/ENCSR386RIJ | Encode portal, 10.17989/ENCSR386RIJ |
| Encode consortium | 2016 | Histone ChIP-seq | https://doi.org/10.17989/ENCSR657DYL | Encode portal, 10.17989/ENCSR657DYL |
| Encode consortium | 2016 | Histone ChIP-seq | https://doi.org/10.17989/ENCSR773IYZ | Encode portal, 10.17989/ENCSR773IYZ |

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
