## [Editor Report]

The work by Vallier and colleagues programmes ESCs and IPSCs towards hepatocyte fate by using a combination of hepatocyte transcription factors. Based on informatic analyses comparing adult hepatocytes with hepatocyte-like cells differentiated with soluble factors they conclude that the inclusion of RORc is important for added maturity of the forward programmed cells. The challenge is a very important one as we still don't have good in vitro hepatocyte generation.

---

## [Decision Letter]

**Decision letter after peer review:**

Thank you for submitting your article "Generation of functional hepatocytes by forward programming with nuclear receptors" for consideration by eLife. Your article has been reviewed by 3 peer reviewers, and the evaluation has been overseen by a Reviewing Editor and Mone Zaidi as the Senior Editor. The reviewers have opted to remain anonymous.

Essential revisions:

The primary revision that is essential is testing the functionality of the induced hepatocytes, specifically, compared to classically induced hepatocytes.

*Reviewer #1 (Recommendations for the authors):*

Directed (classical) differentiation methods to obtain iPSC-derived HLC are time-consuming and lack the full repertoire of functionalities of hepatocytes. This manuscript by Tomaz et al. reports a new method to generate Hepatic Like Cells (HLC) from iPSCs and ESCs. The authors identify a combination of four factors (HNF1A, HNF6, FOXA3, and RORc) that when overexpressed in iPSCs or ESCs induce their programming to HLC. After evaluating functionality, the authors conclude that the forward programmed cells show increased functionality when compared to classic iPSC-derived HLCs. Though the manuscript is well written, involves a large amount of work, and confers a great advance to the field of regenerative medicine, there are several important points to be addressed:

1. Differentiation stability and long-term functionality of iPSC-derived cells are crucial when considering any therapeutic use. In this work, the forward programmed HLCs seem to decline in activity markers (CYP3A4, ALB) over time (see Figure 1 and Fig 4F). This questions whether the programming effect of those 4 factors is reversible. The functionality of 4F forward programmed cells should be assessed beyond day 30 and the functionality should be compared with classical direct programmed HLCs at that timepoint.

2. Are the 4F forward programmed cells functional in vivo?

3. The importance of the findings in this work very much depends on how do the forward programmed hepatic cells (eFoPs and iFoPS) compare to the classic direct reprogrammed HLC (HLCs) and the primary hepatocytes (PHHs). Therefore, those controls (HLCs and PHHs) should be included in all the experiments. i.e Figure 1D-J; Figure 4A-G, Suppl Fig6 should contain HLC and mature hepatocytes as controls.

4. How do the transcriptome of the 3F (HNF1A, HNF6, FOXA2) forward programmed cells and 4F (HNF1A, HNF6, FOXA2, RORc) forward programmed cells compare to the HLCs and PHHs? How does it compare to Foetal hepatocytes?

5. Figure 2 shows there are plenty of pathways differentially activated in PHH when compared to HLCs, which is reflected in plenty of genes including key transcription factors being differentially expressed (Fig.2G). It is not clear why the authors focus their interest exclusively on nuclear receptors. This should be discussed.

6. Figure 5 is redundant to Figure 4. Since this is just another iPSC line, it should either be merged with figure 5 or included as a supplementary figure.

7. Quantification of IF should be done for all the images presented.

8. The statistical analyses are not clear, labelling for p-values should be consistent (either value or *, but consistent in all the graphs). It is unclear at times what groups have been compared in each graph. Comparisons should be clearly indicated (line indicating the 2 groups compared as in Figure 2B). HLC and PHHs should be always included as controls in all the graphs.

9. Fig6a - eFOPS and iFOPS day 30 should be included

10. Fig6B - CYP3A4 gene and ALB should be included here.

*Reviewer #2 (Recommendations for the authors):*

The major strength of the manuscript is the platform on which it is built, namely the system for forward programming and the in house strength of the Vallier group in hepatocyte differentiation. The discovery that three transcription factors, HNF1A, HNF6 and FOXA3, can induce forward programming towards the hepatocyte fate is notable. Equally notable is that HNF4A was dispensable raising questions on the precise phase of HNF4A action during normal development and maturation of hepatocytes. This is interesting and maps to similar questions during in vivo development.

In the informatic analyses I did not quite follow the logic of scrutinising variance by restricting the data to the 500 most variable genes rather than an unbiased scrutiny of the whole dataset. Similarly, the conclusion about H3K27ac modifications being more discriminatory than the H3K27me3 differences was unclear to me. As suggested in my recommendations, I don't think this really matters-the main rationale for studying H3K27ac modifications could be made clearer; namely to arrive at motifs in active regulatory element with potential binding factors that influence hepatocyte maturation. That alone provides a strong case.

While the initial findings on the three TFs are very interesting, I am less convinced about the additional role of RORc from the data presented. Some results between ESC and IPSC derived cells were discordant, differences in effect were apparent between the two later time-points of culture, and some statistical analyses seemed insignificant but accompanied by positive text statements. The main 'take home' message for me was that the resultant cells possess hepatocyte-like qualities but still fall far short of primary adult hepatocytes in terms of their primary function for xenobiotic metabolism. This is not really criticism as that goal is a massive one that will require multiple stepwise breakthroughs. In this regard, while I feel that the Results text needs moderation and perhaps reconsideration in places, the Discussion is very nicely balanced.

I have made my comments in order, rather than priority.

1. Introduction: 'after birth when it takes nearly 12 months for the liver to become functional'; this should be changed. The liver is unequivocally functional from early embryogenesis. No liver equals embryonic lethality.

2. In figure 1G-J the '-' sign gets lost on the x-axis. I suggest 'no' would be clearer.

3. Line 127: I think this should be changed. Measuring mRNA for ALB, SERPINA1, AFP and luciferase based CYP3A4 activity is useful screening but not assessing functional characteristics of hepatocytes in anything like their totality. HNF4A is proven to be a fundamental regulator of the hepatocyte phenotype. While it might be dispensable for some aspects, and may lower the handful of parameters selected, that does not equate to 'functional characteristics of hepatocytes'. I also think the conclusion (line 129) should be forward programming 'towards' rather than 'into' based on the very limited screening analyses undertaken to this point.

4. Figure 1, supplement 2B: it is striking how heterogeneous the albumin detection is (compared to A1AT or AFP)? Compared to the DAPI it is in only a fairly small minority of the cells. I would comment on this. It is not a criticism; we notice the same in vivo. We tend to treat 'hepatocytes' as one population with the potential the varying degrees of differentiation when actually I think they represent quite different subpopulations over and above what we know about zonation.

5. Line 137, I would describe this as 'following birth'. 'Adult' function is not established in the 'neonatal' period (by definition).

6. Line 141: I am not sure what is intended by 'are likely to be less relevant for natural development'?

7. Line 158: I don't understand the logic of scrutinising variance by restricting the choice to the 500 most variable genes. Isn't the point to scrutinise variance within the whole dataset? Restriction to the 500 most variable means the downstream analyses and individual GO terms end up based on remarkably few genes (which should be compiled as supplementary tables). What do the data look like without this arbitrary cut-off? Why 500?

8. Line 192-5. Why is H3K27ac providing the 'strongest distinction' (not a particularly scientific phrase? Variance?) and 'most informative'. There looks to be as much segregation of the PHH replicates on the H3K27me3 plot? I.e. cell identity is as much determined by what is actively repressed as what is actively 'switched on'.

9. Fig. 3 supplement 1C. I am not sure the terms are particularly 'developmental'; more seem to relate to neural function?

10. Fig. 3C right hands labels are cropped; I am not sure what message these panels are trying to convey? The text also becomes hard to follow in line 208-the H3K27ac regions can also harbour K27me3 marks? Line 211-212: I don't see evidence to support the statement, 'taken together these observations suggested that HLCs and PHHs broadly share the same epigenetic identity'-compared to what? In line 214, an association is taken as proof of causality. It potentially explains why these genes are not expressed. This whole section could do with reconsideration. I would favour a re-write with a far simpler focus: we wanted to identify the regulatory regions responsible for the differential gene expression so we could identify the underlying motifs and nuclear receptors/TFs that might bind to them.

11. Nuclear receptors for sex steroids: were the PHH samples from males or females (given the presence of AR and ER motifs)? Do the authors have a comment on the detection of HNF4A (dispensable) but not HNF1A, HNF6 or FOXA3 in the heatmap in Fig 3 supplement 1E? And might the upregulation of AR and ER in mice at 8 and 10 weeks simply reflect post-pubertal status rather than hepatocyte maturity?

12. Figure 4, supplement 1B: I disagree with the authors a little here. AR clearly has a cytoplasmic distribution in addition to potential nuclear detection (it is hard to say given the potential for piled up cells and signal overlying the nucleus that is not actually nuclear; in any case the nature of the AR panel is obviously different compared to the other nuclear receptors). This implies strongly that its ligand, either testosterone or dihydrotestosterone, is absent from the conditions in which case one wouldn't anticipate much functional effect even if AR was important in hepatocytes. Have the authors tried adding the relevant sex steroid to the culture conditions and repeating the immunocytochemistry? For AR it would induce comprehensive nuclear localisation. I disagree about the statement about potentially bypassing the need for ligand. The conformational change induced by ligand binding is highly relevant to DNA binding and gene activation. Just because the relevant receptor may (or may not) be in the nucleus doesn't allow confident statements on activity.

13. Consistency of the data and interpretation / risk of over-interpretation. RORc on albumin. I think the statement made in line 241-2, needs toning down. There isn't a statistical difference? Day 30 ALB mRNA levels are directly equivalent between 3TFs and 3TFs+RORc (Fig. 4D). Arguably (presuming gain settings are the same in panel B), while there are the same number of cells with robust detection of albumin there are more cells with faint detection in the presence of RORc. Likewise, I don't think much can be said about AFP levels being lower given the overlap in range and lack of statistical significance.

14. Consistency of the data and interpretation / risk of over-interpretation. Developing the second cell line: in the text the authors say that RORc overexpression increased expression of AFP and SERPINA1-this is not evident at Day 30 for transcript (D) or protein (E) and is non-significant for albumin for all except the day 20 mRNA analysis. Presumably day 30 is the most mature stage of culture (Fig . 5D and E)? Moreover, in the IPSC analysis RORc is quoted as decreasing AFP transcript detection (Fig. 4D, albeit non-significant), whereas in the second line studied, AFP transcript detection is actually increased by RORc (Figure 5D)-this discrepancy needs explaining as it might legitimately raise concerns that the detected differences are not meaningful.

15. How do the authors explain the lack of effect from the ligand for RORc (Fig. 4G)?

16. Figure 6A. Rather than being a side-by-side analysis, the CYP3A4 analysis (Fig. 6A) looks like it re-uses the preceding eFoP (Fig. 4A) and iFoP data (Fig. 5F) (the position of the balls and the SD bars look the same?). However, the HLC v PHH data look different from those in Figure 2B when the P value for their difference was only 0.0190. Here, P is 0.0002 for eFoP v. HLC? Given the massively higher activity in PHH, this suggests the HLCs in Figure 6 are suboptimal compared to those in Figure 2. For robustness, these comparative experiments ought to be done side-by-side. For me, the main message is that I agree with the authors that the considerable 'ask' remains unmet for CYP expression compared to PHH.

*Reviewer #3 (Recommendations for the authors):*

Valier's lab is one of the pioneering labs in endoderm differentiation and generation of hepatocyte- and cholangiocyte-like cells from human pluripotent stem cells (hPSCs) and has always performed and published high-quality research. This paper is no different in that respect. The authors worked meticulously and systematically to find the right combination of transcription factors involved in the liver development for forward programming of hPSCs. Comparing the transcriptome profile of primary human hepatocytes (PHHs) and hepatocyte-like cells (HLCs), they identified a number of nuclear receptors known to play a role in liver metabolic activity. To further improve the functionality of forward programmed HLCs, the authors tested the induction of three nuclear receptors in combination with three liver-enriched transcription factors. Notably, overexpression of RORc was found to reduced alpha-fetoprotein secretion as a foetal marker and improved Albumin protein production and CYP3A4 metabolic activity.

The conclusions of this paper are supported mainly by data, as forward programming can be considered a less complicated and more cost-effective alternative methodology than stepwise differentiation of hPSCs into HLCs using recombinant growth factors. However, it would be helpful to clarify the following points to draw a better conclusion:

1) Since the study aimed to generate HLCs with improved functionalities, comparing the level of HNF6, FOXA3 and HNF1 between generated forward programmed HLCs (FoP-Heps) and PHHs, considering that doxycycline (dox) induction was maintained throughout the differentiation.

2) The authors indicated that culture medium supplemented with 1mg/ml dox for the protocol duration (page 19, line 443). If tested, it would be helpful to share the consequence of dox exclusion after cells acquired hepatic phenotype. Would the FoP-Heps dedifferentiate and lose hepatic phenotype earlier than usual? If yes, then it would be helpful to elaborate on the potential implication for using the FoP-Heps in disease modelling and drug screening in the discussion.

3) While gene expression analysis for both FoP-Heps and HLCs were included for some genes (Figure 6 and Figure 6 Supplement 1), comparison of key hepatic functions such as secretion of ALB, AFP and Alpha1-Antitrypsin and CYP3A4 metabolic functionality are missing. The addition of these data would help to make a direct comparison between HLCs and FoP-Heps functionality.

4) It is unclear how long 2D-cultured FoP-Heps maintain their hepatic phenotype and function in culture.

The forward programming is an intriguing idea, and I would like to congratulate the authors for performing and presenting the data at such a high standard which is not surprising considering the high calibre of the Principal Investigator.

---

## [Author Response]

Essential revisions:The primary revision that is essential is testing the functionality of the induced hepatocytes, specifically, compared to classically induced hepatocytes.Reviewer #1 (Recommendations for the authors):Directed (classical) differentiation methods to obtain iPSC-derived HLC are time-consuming and lack the full repertoire of functionalities of hepatocytes. This manuscript by Tomaz et al. reports a new method to generate Hepatic Like Cells (HLC) from iPSCs and ESCs. The authors identify a combination of four factors (HNF1A, HNF6, FOXA3, and RORc) that when overexpressed in iPSCs or ESCs induce their programming to HLC. After evaluating functionality, the authors conclude that the forward programmed cells show increased functionality when compared to classic iPSC-derived HLCs. Though the manuscript is well written, involves a large amount of work, and confers a great advance to the field of regenerative medicine, there are several important points to be addressed:1. Differentiation stability and long-term functionality of iPSC-derived cells are crucial when considering any therapeutic use. In this work, the forward programmed HLCs seem to decline in activity markers (CYP3A4, ALB) over time (see Figure 1 and Fig 4F). This questions whether the programming effect of those 4 factors is reversible. The functionality of 4F forward programmed cells should be assessed beyond day 30 and the functionality should be compared with classical direct programmed HLCs at that timepoint.

We acknowledge that stability and long-term functionality are key requirements that need to be met by stem-cell derived cell types in order to be applicable in therapy. Indeed FoP-Heps display higher levels of CYP3A4 activity at day 20 and these appear to reduce by day 30. However, hepatocyte marker expression in FoP-Heps did not appear to decreased from day 20 to day 30, apart from AFP (Figure 4D). We have previously characterised direct differentiation HLCs after long term cultures and found that conventional 2D culture systems were not compatible with maintaining HLCs functionality beyond day 45, resulting in cell detachment, potentiality due to lack of an appropriate matrix. Instead, it was found that 3D systems, such as the collagen-based RATF system, were appropriate to preserve an hepatic phenotype and further increase functionality up to 75 days (Gieseck et al, 2014).

In the current study, we have developed a forward programming method to generate hepatocytes using conventional 2D cultures which we believe limits their applicability and longterm functionality. We have differentiated FoP-Heps up to day 40 and observed that these still retain CYP3A4 activity, as well as HLCs (Author response image 1), but the cell attachment tends to deteriorate with longer times in culture (not shown). Here, we aimed to develop a versatile and faster alternative to generate hepatocytes in vitro and have chosen to characterised the generated cells in their optimal time frame (day 20). Nevertheless, we agree with the reviewer that should this method be further validated for therapeutic use, improvements should be made to extend their lifespan and usability, potentially adapting extra cellular matrix to meet the requirements for proper hepatocyte attachment and/or 3D system.

**Author response image 1. sa2fig1:** CYP3A4 Activity in long-term cultures. hiPSCs expressing the 4TF combination were induced to differentiate for up to 40 days, with CYP3A4 activity measure at days 20, 30 and 40 (left panel). HLCs where differentiated by directed differentiation up to 40 days, with CYP3A4 activity measured at days 30, 35 and 40 (right panel). Data is represented as fold induction over day 20 (iFoP) or day 30 (HLCs).

2. Are the 4F forward programmed cells functional in vivo?

Assessing in vivo functionality is key to define the potential of forward programmed cells. Indeed, we have deployed our resources into addressed this question during the course of the revision by performing in vivo injections of eFoPs and iFoPs using a healthy mouse model. Initially, we performed injections in kidney capsule but this approach showed no hAlbumin detection on mice serum by up to 4 weeks (Author response image 2). Subsequently, we performed injections directly into the liver capsule which was found to provide a more appropriate niche, as hAlbumin expression could indeed be detected in mice serum at 3 or 4 weeks post-injection (Author response image 2), although at variable levels. However, hAlbumin was not detected consistently in all mice injected highlighting that the use of a healthy mouse model was inappropriate for sufficient and robust integration of the forward programmed cells into the liver. Using a liver injury model would provide a more suitable environment, however the time frame required to establish these models or to establish appropriate collaborations would fall beyond the appropriate time for this revision, and thus we propose to pursue these experiments as a follow-up study with suitable mouse models.

**Author response image 2. sa2fig2:** Human Albumin secretion in mice sera. eFoP (hESC-derived) or iFoP (hIPSC-derived) cells differentiated up to 20 days were dissociated into small clusters, and 1 million cells mixed with growth-factor reduced Matrigel and injected (per mouse) into the kidney or liver capsules of healthy mice. Blood samples were collected at the timepoints indicated post-injection.

3. The importance of the findings in this work very much depends on how do the forward programmed hepatic cells (eFoPs and iFoPS) compare to the classic direct reprogrammed HLC (HLCs) and the primary hepatocytes (PHHs). Therefore, those controls (HLCs and PHHs) should be included in all the experiments. i.e Figure 1D-J; Figure 4A-G, Suppl Fig6 should contain HLC and mature hepatocytes as controls.

Data for the controls HLCs and PHHs has now been included in every figure for comparison where relevant, in the revised version of this manuscript.

4. How do the transcriptome of the 3F (HNF1A, HNF6, FOXA2) forward programmed cells and 4F (HNF1A, HNF6, FOXA2, RORc) forward programmed cells compare to the HLCs and PHHs? How does it compare to Foetal hepatocytes?

We have performed RNA-sequencing analysis on newly generated datasets of eFoP (4TFs) and foetal liver samples, in order to characterise what we identified as cells generated with the best combination of TFs identified in our study (4TFs). This data has been incorporated into a new Figure 5 and Figure 5 – supplement 2. From these analysis, eFoP cells seem to be distinct from foetal liver cells, as well as adult PHHs. eFoP were found to cluster in close proximity to HLCs derived from hESC and hiPSCs, and thus reinforce these to be an equivalent cell population generated with an alternative method. Interestingly, eFoP express adult liver genes similarly to HLCs, including genes not yet expressed in foetal liver cells.

5. Figure 2 shows there are plenty of pathways differentially activated in PHH when compared to HLCs, which is reflected in plenty of genes including key transcription factors being differentially expressed (Fig.2G). It is not clear why the authors focus their interest exclusively on nuclear receptors. This should be discussed.

Indeed, several pathways are lacking activation in HLCs when compared to PHHs and our differential gene expression analysis identified 36 TFs highly expressed in PHHs vs HLCs (p < 0.05, log2 fold change > 2). It is possible that many of the TFs identified in this study play a key role in hepatocyte maturation. However, it is known that nuclear receptors are involved in key liver functions including the metabolism of lipid and glucose levels, bile acid clearance, xenobiotic sensing and regeneration (Rudraiah et al., 2016). Given that the approach that we used to test the role of TFs in hepatocyte maturation involved gene targeting, generation of cell lines, and differentiation, all of which are lengthy processes, we decided to narrow down the list of candidate TFs for our study to nuclear receptors, since based on literature there was a high chance that these would could improve hepatocyte metabolism. The choice for nuclear receptors has been discussed in the results section in the revised version of this manuscript (line 174).

6. Figure 5 is redundant to Figure 4. Since this is just another iPSC line, it should either be merged with figure 5 or included as a supplementary figure.

Figures 5 has been relabelled as Figure 4 – figure supplement 2. In the revised version of this manuscript.

7. Quantification of IF should be done for all the images presented.

The authors agree that quantification is essential, especially for comparisons among cell lines overexpressing different TF combinations. Indeed we provide quantifiable data in qPCR and ELISA for all the same markers shown in IF as we believe these to be more robust and reliable methods for quantification and not dependent of staining/image quality or field of imaging.

8. The statistical analyses are not clear, labelling for p-values should be consistent (either value or *, but consistent in all the graphs). It is unclear at times what groups have been compared in each graph. Comparisons should be clearly indicated (line indicating the 2 groups compared as in Figure 2B). HLC and PHHs should be always included as controls in all the graphs.

For consistency, all figures have been amended to identify the groups compared in each test for which a significant p-value was obtained. P-value groups for all figures have been defined as: *p < 0.05, **p < 0.01, ***p< 0.001, ****p< 0.0001. HLCs and PHHs have been included as controls in every graph.

9. Fig6a - eFOPS and iFOPS day 30 should be included

CYP3A4 activity measurements taken on day 30 have been included for eFoP and iFoP in the new Figure 5A (previously named 6A), in the revised version of this manuscript.

10. Fig6B - CYP3A4 gene and ALB should be included here.

Data for CYP3A4 and Albumin has now been included in the new Figure 5B and C (previously named 6A and 6B), respectively, in the revised version of this manuscript.

Reviewer #2 (Recommendations for the authors):The major strength of the manuscript is the platform on which it is built, namely the system for forward programming and the in house strength of the Vallier group in hepatocyte differentiation. The discovery that three transcription factors, HNF1A, HNF6 and FOXA3, can induce forward programming towards the hepatocyte fate is notable. Equally notable is that HNF4A was dispensable raising questions on the precise phase of HNF4A action during normal development and maturation of hepatocytes. This is interesting and maps to similar questions during in vivo development.In the informatic analyses I did not quite follow the logic of scrutinising variance by restricting the data to the 500 most variable genes rather than an unbiased scrutiny of the whole dataset. Similarly, the conclusion about H3K27ac modifications being more discriminatory than the H3K27me3 differences was unclear to me. As suggested in my recommendations, I don't think this really matters-the main rationale for studying H3K27ac modifications could be made clearer; namely to arrive at motifs in active regulatory element with potential binding factors that influence hepatocyte maturation. That alone provides a strong case.While the initial findings on the three TFs are very interesting, I am less convinced about the additional role of RORc from the data presented. Some results between ESC and IPSC derived cells were discordant, differences in effect were apparent between the two later time-points of culture, and some statistical analyses seemed insignificant but accompanied by positive text statements. The main 'take home' message for me was that the resultant cells possess hepatocyte-like qualities but still fall far short of primary adult hepatocytes in terms of their primary function for xenobiotic metabolism. This is not really criticism as that goal is a massive one that will require multiple stepwise breakthroughs. In this regard, while I feel that the Results text needs moderation and perhaps reconsideration in places, the Discussion is very nicely balanced.I have made my comments in order, rather than priority.1. Introduction: 'after birth when it takes nearly 12 months for the liver to become functional'; this should be changed. The liver is unequivocally functional from early embryogenesis. No liver equals embryonic lethality.

The sentence had been modified accordingly in the revised version of this manuscript (line 57).

2. In figure 1G-J the '-' sign gets lost on the x-axis. I suggest 'no' would be clearer.

The axis on Fig.1G-J have been modified accordingly in the revised version of this manuscript as well as every figure where “-“ was previously used. .

3. Line 127: I think this should be changed. Measuring mRNA for ALB, SERPINA1, AFP and luciferase based CYP3A4 activity is useful screening but not assessing functional characteristics of hepatocytes in anything like their totality. HNF4A is proven to be a fundamental regulator of the hepatocyte phenotype. While it might be dispensable for some aspects, and may lower the handful of parameters selected, that does not equate to 'functional characteristics of hepatocytes'. I also think the conclusion (line 129) should be forward programming 'towards' rather than 'into' based on the very limited screening analyses undertaken to this point.

The text has been modified accordingly in the revised version of this manuscript (line 127 and line 127).

4. Figure 1, supplement 2B: it is striking how heterogeneous the albumin detection is (compared to A1AT or AFP)? Compared to the DAPI it is in only a fairly small minority of the cells. I would comment on this. It is not a criticism; we notice the same in vivo. We tend to treat 'hepatocytes' as one population with the potential the varying degrees of differentiation when actually I think they represent quite different subpopulations over and above what we know about zonation.

We acknowledge that indeed there is heterogeneity in the Albumin levels which can be detected with all TF combinations, and could relate to subpopulations. We have commented on the heterogeneity displayed by Albumin expression in the revised version of this manuscript (line 121 and line 244).

5. Line 137, I would describe this as 'following birth'. 'Adult' function is not established in the 'neonatal' period (by definition).

The text has been modified accordingly in the revised version of this manuscript (line 137).

6. Line 141: I am not sure what is intended by 'are likely to be less relevant for natural development'?

FoP-heps are not differentiated by mimicking different steps of liver development, like the process for directed differentiationto generate HLCs. Moreover, HLCs have been previously been reported to represent a foetal stage (Baxter et al., 2015). Thus, we assumed HLCs from directed differentiation would serve as natural “immature” state of hepatocytes as control for the RNA-seq comparison of immature vs mature hepatocytes.

7. Line 158: I don't understand the logic of scrutinising variance by restricting the choice to the 500 most variable genes. Isn't the point to scrutinise variance within the whole dataset? Restriction to the 500 most variable means the downstream analyses and individual GO terms end up based on remarkably few genes (which should be compiled as supplementary tables). What do the data look like without this arbitrary cut-off? Why 500?

We appreciate the reviewer’s remark on the choice of 500 most variable genes for a principal component analysis. This is a default parameter of the PCA function, and it is used to focus the PCA plot on the most variable genes across different samples, which will most likely relate to the differential expression identified between sample groups. We have performed PCA using 1000, 5000 and 20.000 genes and did not observe any differences in the clustering of the samples (Author response image 3), thus we assume that the top 500 most variable genes sufficiently captures the differences and similarities between samples.

**Author response image 3. sa2fig3:** PCA plot. PCA of undifferentiated hiPSCs, HLCs derived from hESC (hESC_HLCs) and hiPSC (hiPSC_HLCs), freshly harvested PHHs (fPHHs) or plated PHHs (pPHHs), where generated with the top 1000 (left panel), 5000 (middle panel) and 20.000 (right panel) most variable genes.

8. Line 192-5. Why is H3K27ac providing the 'strongest distinction' (not a particularly scientific phrase? Variance?) and 'most informative'. There looks to be as much segregation of the PHH replicates on the H3K27me3 plot? I.e. cell identity is as much determined by what is actively repressed as what is actively 'switched on'.

The text has been modified accordingly in the revised version of this manuscript (line 195).

9. Fig. 3 supplement 1C. I am not sure the terms are particularly 'developmental'; more seem to relate to neural function?

The text has been modified accordingly in the revised version of this manuscript (line 204).

10. Fig. 3C right hands labels are cropped; I am not sure what message these panels are trying to convey? The text also becomes hard to follow in line 208-the H3K27ac regions can also harbour K27me3 marks? Line 211-212: I don't see evidence to support the statement, 'taken together these observations suggested that HLCs and PHHs broadly share the same epigenetic identity'-compared to what? In line 214, an association is taken as proof of causality. It potentially explains why these genes are not expressed. This whole section could do with reconsideration. I would favour a re-write with a far simpler focus: we wanted to identify the regulatory regions responsible for the differential gene expression so we could identify the underlying motifs and nuclear receptors/TFs that might bind to them.

Figure 3C has been modified to have visible panels which show the levels of all histone modifications across the UGT1A locus. The text has been clarified to convey the aim of the epigenetic analysis, as suggested by the reviewer (line 187, 215, 217).

11. Nuclear receptors for sex steroids: were the PHH samples from males or females (given the presence of AR and ER motifs)? Do the authors have a comment on the detection of HNF4A (dispensable) but not HNF1A, HNF6 or FOXA3 in the heatmap in Fig 3 supplement 1E? And might the upregulation of AR and ER in mice at 8 and 10 weeks simply reflect post-pubertal status rather than hepatocyte maturity?

We appreciate the reviewer’s questions on nuclear receptors for sex steroids. Both proteins have been reported to be expressed in both genders in rodents and humans. In addition, their expression is age-dependent in rodents, being expressed at higher levels post-puberty (Shen and Shi, 2015). Indeed, we cannot establish causality between expression or AR and ERa and hepatic maturation, but the observation that these are specifically expressed in adult / postpubertal livers, versus foetal stage livers, was interesting as it reiterates that these markers seem to be “adult-specific”, as suggested by our PHH /HLCs RNA-seq comparisons. The PHH samples used in the ChIP-sequencing analyses were derived from a male donor. In our PHH qPCR expression data which is limited to 3 male and 1 female donor, we could not identify a clear difference in expression levels of these markers associated with the gender of the donor (Author response image 4).

The identification of *HNF4A* specifically in mouse liver RNA-seq was due to this heatmap specifically representing transcription factors that are categorised as nuclear receptors, and hence *FOXA3, HNF6, HNF1A* would not be expected to appear in this analysis in particular as they are not categorised as such.

**Author response image 4. sa2fig4:** Expression of nuclear receptors in PHHs. mRNA levels of *AR* and *ESR1* (ERa) genes in primary human hepatocytes from 4 donors including 1 female (F) and 3 males (M1, M2, M3). Data were normalised to the average of 2 housekeeping genes (*PBGD* and *RPLP0*).

12. Figure 4, supplement 1B: I disagree with the authors a little here. AR clearly has a cytoplasmic distribution in addition to potential nuclear detection (it is hard to say given the potential for piled up cells and signal overlying the nucleus that is not actually nuclear; in any case the nature of the AR panel is obviously different compared to the other nuclear receptors). This implies strongly that its ligand, either testosterone or dihydrotestosterone, is absent from the conditions in which case one wouldn't anticipate much functional effect even if AR was important in hepatocytes. Have the authors tried adding the relevant sex steroid to the culture conditions and repeating the immunocytochemistry? For AR it would induce comprehensive nuclear localisation. I disagree about the statement about potentially bypassing the need for ligand. The conformational change induced by ligand binding is highly relevant to DNA binding and gene activation. Just because the relevant receptor may (or may not) be in the nucleus doesn't allow confident statements on activity.

We agree with the reviewer’s comment that we cannot assume that the overexpression system used bypasses the need for a ligand. Indeed, we acknowledge that the pattern of staining of AR is clearly different from the other nuclear receptors RORc and ERa which are uniquely nuclear. Adding the ligand to the culture conditions would potentially induce comprehensive translocation to the nucleus. However, from the immunofluorescence results it appears that AR is present in both nucleus and cytoplasm in our overexpressing cell line without ligand, and thus we would expect to have some level transcriptional activity. We present here a different biological replicate of this staining with a closer field that better represents the dual localisation of this protein in the cytoplasm and nucleus (Author response image 5). We have modified the text to remove statements regarding activity related to AR overexpression (line 238).

**Author response image 5. sa2fig5:** Cellular localisation of AR. Immunofluorescence staining of HNF1A an AR in hESCs targeted with 3TFs+AR after 24h of iOX with dox. Scale bar, 100µm.

13. Consistency of the data and interpretation / risk of over-interpretation. RORc on albumin. I think the statement made in line 241-2, needs toning down. There isn't a statistical difference? Day 30 ALB mRNA levels are directly equivalent between 3TFs and 3TFs+RORc (Fig. 4D). Arguably (presuming gain settings are the same in panel B), while there are the same number of cells with robust detection of albumin there are more cells with faint detection in the presence of RORc. Likewise, I don't think much can be said about AFP levels being lower given the overlap in range and lack of statistical significance.

We agree with the reviewer that the effect in Albumin and AFP is not statistically significant and instead it shows simply is a trend in the secretion of these two proteins, not supported by changes in transcript levels. We have modified the text to describe this data accurately (line 246).

14. Consistency of the data and interpretation / risk of over-interpretation. Developing the second cell line: in the text the authors say that RORc overexpression increased expression of AFP and SERPINA1-this is not evident at Day 30 for transcript (D) or protein (E) and is non-significant for albumin for all except the day 20 mRNA analysis. Presumably day 30 is the most mature stage of culture (Fig . 5D and E)? Moreover, in the IPSC analysis RORc is quoted as decreasing AFP transcript detection (Fig. 4D, albeit non-significant), whereas in the second line studied, AFP transcript detection is actually increased by RORc (Figure 5D)-this discrepancy needs explaining as it might legitimately raise concerns that the detected differences are not meaningful.

We have modified the text to describe this data accurately. Indeed, for the hiPSC background, it appears that cells continue to induce ALB expression by day 30, suggesting they require additional time for maturation (line 268).

15. How do the authors explain the lack of effect from the ligand for RORc (Fig. 4G)?

Intermediates of the cholesterol biosynthetic pathway have been found to bind to and enhance the transcriptional activity of *RORy* (RORc), and these have been proposed to act as endogenous ligands. In particular, the cholesterol precursor desmosterol has been reported as one of most effective ligands and thus it has been selected as a candidate ligand in this study. However, most studies investigating the potential of endogenous ligands have focused on the activity of *RORyt*, an isoform which is specifically expressed in a range of immune cells including T helper cells (Th7). Indeed, the effect of sterols in the differentiation of Th7, via transcriptional activation of *RORyt*, has been widely studied, but it is unclear whether the same sterols have the same effect on the *RORy* isoform expressed in the liver. In addition, it is possible that the FoP-heps activate to some extend the cholesterol biosynthetic pathway and provide endogenous ligands that activate the transcriptional activity of RORc.

16. Figure 6A. Rather than being a side-by-side analysis, the CYP3A4 analysis (Fig. 6A) looks like it re-uses the preceding eFoP (Fig. 4A) and iFoP data (Fig. 5F) (the position of the balls and the SD bars look the same?). However, the HLC v PHH data look different from those in Figure 2B when the P value for their difference was only 0.0190. Here, P is 0.0002 for eFoP v. HLC? Given the massively higher activity in PHH, this suggests the HLCs in Figure 6 are suboptimal compared to those in Figure 2. For robustness, these comparative experiments ought to be done side-by-side. For me, the main message is that I agree with the authors that the considerable 'ask' remains unmet for CYP expression compared to PHH.

The HLC and PHH datasets used (including both CYP3A4 data and expression data) are the same throughout the figures and are now included in every figure for comparison as suggested. Previously, HLC vs PHH CYP3A4 data was presented as log scale when compared alone in Figure 2 and linear scale when compared with FoP in Figure 6. We have now modified the figures so that all data is presented as linear scale and the reader can best identified when the same data is presented in multiple Figures.

Figure 6 (now renamed as Figure 5) aims to compare data from 4TF lines from both backgrounds (eFoP and iFoP) with controls and reused data from 4TF experiments in Figure 4 and 5 (now renamed as Figure 4 and supplement). We agree with the reviewer that ideally all experiments in different figures should be performed “side-by-side”, but given the multiple cell lines and replicates used throughout this study for 20/30-day long protocols we found this to be technically challenging. Thus, we have performed side-by-side experiments only in the case where different TF combinations where screened for the same cell line background: 3TF combinations in Figure 1G-J, 4TF combinations in Figure D-F, and 3TF vs 4TF in Figure 4supplement 2 (previously Figure 5).

We indicate in the source data every time the same dataset is used as control in multiple figures.

Reviewer #3 (Recommendations for the authors):Valier's lab is one of the pioneering labs in endoderm differentiation and generation of hepatocyte- and cholangiocyte-like cells from human pluripotent stem cells (hPSCs) and has always performed and published high-quality research. This paper is no different in that respect. The authors worked meticulously and systematically to find the right combination of transcription factors involved in the liver development for forward programming of hPSCs. Comparing the transcriptome profile of primary human hepatocytes (PHHs) and hepatocyte-like cells (HLCs), they identified a number of nuclear receptors known to play a role in liver metabolic activity. To further improve the functionality of forward programmed HLCs, the authors tested the induction of three nuclear receptors in combination with three liver-enriched transcription factors. Notably, overexpression of RORc was found to reduced alpha-fetoprotein secretion as a foetal marker and improved Albumin protein production and CYP3A4 metabolic activity.The conclusions of this paper are supported mainly by data, as forward programming can be considered a less complicated and more cost-effective alternative methodology than stepwise differentiation of hPSCs into HLCs using recombinant growth factors. However, it would be helpful to clarify the following points to draw a better conclusion:1) Since the study aimed to generate HLCs with improved functionalities, comparing the level of HNF6, FOXA3 and HNF1 between generated forward programmed HLCs (FoP-Heps) and PHHs, considering that doxycycline (dox) induction was maintained throughout the differentiation.

We have included the expression data for *HNF1A*, *FOXA3* and *HNF6* in the new Figure 5 – supplement 1.

2) The authors indicated that culture medium supplemented with 1mg/ml dox for the protocol duration (page 19, line 443). If tested, it would be helpful to share the consequence of dox exclusion after cells acquired hepatic phenotype. Would the FoP-Heps dedifferentiate and lose hepatic phenotype earlier than usual? If yes, then it would be helpful to elaborate on the potential implication for using the FoP-Heps in disease modelling and drug screening in the discussion.

We appreciate the reviewer’s comment and agree it would be important to validate whether doxycycline is required for the whole duration of the differentiation or only required for the initial induction of transcription factors. We have indeed performed the 20-day forward programming protocol with dox removal from day 10 and day 15 onwards, and have observed no difference in cell morphology (not shown), as well as CYP3A4 activity and hepatocyte marker expression (Author response image 6). This suggests that actually dox is only necessary for initial induction of the transgenes and to initiate cell commitment, which will likely lead to the expression of the endogenous transcription factors. We consider this finding very relevant for the applicability of this method into clinic as it shows the phenotype is stable even in the absence of long-term doxycycline treatment. Therefore, we have included this information in the discussion section (line 404) and we propose that a further refinement of the current forward programming protocol should be presented a follow-up study.

**Author response image 6. sa2fig6:** Dox removal during forward programming. hiPSCs overexpressing the 4TF combination were induced to differentiate for 20 days. Dox was removed from 10 and from day 15. iFoP-Heps where assessed for CYP3A4 activity levels (left panel). Expression levels of *ALB*, *SERPINA1* and *CYP2A6* were also quantified and normalised to the average of 2 housekeeping genes (*PBGD* and *RPLP0*). Data is represented as fold induction over control (continuous dox over 20 days).

3) While gene expression analysis for both FoP-Heps and HLCs were included for some genes (Figure 6 and Figure 6 Supplement 1), comparison of key hepatic functions such as secretion of ALB, AFP and Alpha1-Antitrypsin and CYP3A4 metabolic functionality are missing. The addition of these data would help to make a direct comparison between HLCs and FoP-Heps functionality.

We agree with the reviewer that including HLC and PHH data as controls throughout the manuscript is helpful for direct comparison. Indeed, we have now included the datasets for CYP3A4 activity on HLCs and PHH with every FoP-Heps dataset. In addition, we have included the expression data for HLCs and PHH as controls in every graph with FoP-Heps data, including quantification of expression of *Alb*, *AFP* and *SERPINA1*. We indicate in the source data every time the same dataset is used as control in multiple figures.

4) It is unclear how long 2D-cultured FoP-Heps maintain their hepatic phenotype and function in culture.

We appreciate the reviewer’s question about long-term stability. The forward programming method in this study relies on conventional 2D cultures which we believe limits their applicability and long-term functionality, which we have experienced with direct differentiation, possibly due to lack of an appropriate matrix to sustain long-term culture.

We have differentiated FoP-Heps up to day 40 and observed that these retain CYP3A4 activity, (Author response image 1 and Author response image 7), but the cell attachment deteriorates with longer times in culture (not shown). Here, we aimed to develop a versatile and faster alternative to generate hepatocytes in vitro and have chosen to characterised the generated cells in their optimal time frame (day 20). Nevertheless, we agree with the reviewer that should this method be further validated for therapy, improvements should be made to extend their lifespan and usability, possibly with appropriate liver ECM and/or using 3D systems. We propose that a further refinement of the current forward programming protocol should be presented a followup study

**Author response image 7. sa2fig7:** CYP3A4 Activity in long term cultures. hiPSCs expressing the 4TF combination were induced to differentiate for up to 40 days, with CYP3A4 activity measured at days 20, 30 and 40.